# Offline Reinforcement Learning via High-Fidelity Generative Behavior Modeling

**Huayu Chen[1], Cheng Lu[1], Chengyang Ying[1], Hang Su[1,2]\*, Jun Zhu[1,2]\***
[1]Department of Computer Science & Technology, Institute for AI, BNRist Center,
Tsinghua-Bosch Joint ML Center, THBI Lab, Tsinghua University
[2]Pazhou Lab, Guangzhou, 510330, China
`chenhuay21@mails.tsinghua.edu.cn`
`{lucheng.lc15,yingcy17}@gmail.com`
`{suhangss,dcszj}@tsinghua.edu.cn`

## Abstract

In offline reinforcement learning, weighted regression is a common method to ensure the learned policy stays close to the behavior policy and to prevent selecting out-of-sample actions. In this work, we show that due to the limited distributional expressivity of policy models, previous methods might still select unseen actions during training, which deviates from their initial motivation. To address this problem, we adopt a generative approach by decoupling the learned policy into two parts: an expressive generative behavior model and an action evaluation model. The key insight is that such decoupling avoids learning an explicitly parameterized policy model with a closed-form expression. Directly learning the behavior policy allows us to leverage existing advances in generative modeling, such as diffusion-based methods, to model diverse behaviors. As for action evaluation, we combine our method with an in-sample planning technique to further avoid selecting out-of-sample actions and increase computational efficiency. Experimental results on D4RL datasets show that our proposed method achieves competitive or superior performance compared with state-of-the-art offline RL methods, especially in complex tasks such as AntMaze. We also empirically demonstrate that our method can successfully learn from a heterogeneous dataset containing multiple distinctive but similarly successful strategies, whereas previous unimodal policies fail. The source code is provided at `https://github.com/ChenDRAG/SfBC`.

## 1 Introduction

Offline reinforcement learning seeks to solve decision-making problems without interacting with the environment. This is compelling because online data collection can be dangerous or expensive in many realistic tasks. However, relying entirely on a static dataset imposes new challenges. One is that policy evaluation is hard because the mismatch between the behavior and the learned policy usually introduces extrapolation error (Fujimoto et al., 2019). In most offline tasks, it is difficult or even impossible for the collected transitions to cover the whole state-action space. When evaluating the current policy via dynamic programming, leveraging actions that are not presented in the dataset (out-of-sample) may lead to highly unreliable results, and thus performance degrade. Consequently, in offline RL it is critical to stay close to the behavior policy during training.

Recent advances in model-free offline methods mainly include two lines of work. The first is the adaptation of existing off-policy algorithms. These methods usually include value pessimism about unseen actions or regulations of feasible action space (Fujimoto et al., 2019; Kumar et al., 2019; 2020). The other line of work (Peng et al., 2019; Wang et al., 2020; Nair et al., 2020) is derived from constrained policy search and mainly trains a parameterized policy via weighted regression. Evaluations of every state-action pair in the dataset are used as regression weights.

The main motivation behind weighted policy regression is that it helps prevent querying out-of-sample actions (Nair et al., 2020; Kostrikov et al., 2022). However, we find that this argument is untenable in

---

*H. Su and J. Zhu are corresponding authors.

certain settings. Our key observation is that policy models in existing weighted policy regression methods are usually unimodal Gaussian models and thus lack distributional expressivity, while in the real world collected behaviors can be highly diverse. This distributional discrepancy might eventually lead to selecting unseen actions. For instance, given a bimodal target distribution, fitting it with a unimodal distribution unavoidably results in covering the low-density area between two peaks. In Section 3.1, we empirically show that lack of policy expressivity may lead to performance degrade.

Ideally, this problem could be solved by switching to a more expressive distribution class. However, it is nontrivial in practice since weighted regression requires exact and derivable density calculation, which places restrictions on distribution classes that we can choose from. Especially, we may not know what the behavior or optimal policy looks like in advance.

To overcome the limited expressivity problem, we propose to decouple the learned policy into two parts: an expressive generative behavior model and an action evaluation model. Such decoupling avoids explicitly learning a policy model whose target distribution is difficult to sample from, whereas learning a behavior model is much easier because sampling from the behavior policy is straightforward given the offline dataset collected by itself. Access to data samples from the target distribution is critical because it allows us to leverage existing advances in generative methods to model diverse behaviors. To sample from the learned policy, we use importance sampling to select actions from candidates proposed by the behavior model with the importance weights computed by the action evaluation model, which we refer to as **S**electing **f**rom **B**ehavior **C**andidates (**SfBC**).

However, the selecting-from-behavior-candidates approach introduces new challenges because it requires modeling behaviors with high fidelity, which directly determines the feasible action space. A prior work (Ghasemipour et al., 2021) finds that typically-used VAEs do not align well with the behavior dataset, and that introducing building-in good inductive biases in the behavior model improves the algorithm performance. Instead, we propose to learn from diverse behaviors using a much more expressive generative modeling method, namely diffusion probabilistic models (Ho et al., 2020), which have recently achieved great success in modeling diverse image distributions, outperforming other existing generative models (Dhariwal & Nichol, 2021). We also propose a planning-based operator for Q-learning, which performs implicit planning strictly within dataset trajectories based on the current policy, and is provably convergent. The planning scheme greatly reduces bootstrapping steps required for dynamic programming and thus can help to further reduce extrapolation error and increase computational efficiency.

The main contributions of this paper are threefold: 1. We address the problem of limited policy expressivity in conventional methods by decoupling policy learning into behavior learning and action evaluation, which allows the policy to inherit distributional expressivity from a diffusion-based behavior model. 2. The learned policy is further combined with an implicit in-sample planning technique to suppress extrapolation error and assist dynamic programming over long horizons. 3. Extensive experiments demonstrate that our method achieves competitive or superior performance compared with state-of-the-art offline RL methods, especially in sparse-reward tasks such as AntMaze.

## 2 BACKGROUND

### 2.1 CONSTRAINED POLICY SEARCH IN OFFLINE RL

Consider a Markov Decision Process (MDP), described by a tuple $\langle \mathcal{S}, \mathcal{A}, P, r, \gamma \rangle$. $\mathcal{S}$ denotes the state space and $\mathcal{A}$ is the action space. $P(\boldsymbol{s}'|\boldsymbol{s}, \boldsymbol{a})$ and $r(\boldsymbol{s}, \boldsymbol{a})$ respectively represent the transition and reward functions, and $\gamma \in (0, 1]$ is the discount factor. Our goal is to maximize the expected discounted return $J(\pi) = \mathbb{E}_{\boldsymbol{s} \sim \rho_\pi(\boldsymbol{s})} \mathbb{E}_{\boldsymbol{a} \sim \pi(\cdot|\boldsymbol{s})} [r(\boldsymbol{s}, \boldsymbol{a})]$ of policy $\pi$, where $\rho_\pi(\boldsymbol{s}) = \sum_{n=0}^{\infty} \gamma^n p_\pi(\boldsymbol{s}_n = \boldsymbol{s})$ is the discounted state visitation frequencies induced by the policy $\pi$ (Sutton & Barto, 1998).

According to the *policy gradient theorem* (Sutton et al., 1999), given a parameterized policy $\pi_\theta$, and the policy's state-action function $Q^\pi$, the gradient of $J(\pi_\theta)$ can be derived as:

$$\nabla_\theta J(\pi_\theta) = \int_{\mathcal{S}} \rho_\pi(\boldsymbol{s}) \int_{\mathcal{A}} \nabla_\theta \pi_\theta(\boldsymbol{a}|\boldsymbol{s}) Q^\pi(\boldsymbol{s}, \boldsymbol{a}) \mathrm{d}\boldsymbol{a} \, \mathrm{d}\boldsymbol{s}. \tag{1}$$

When online data collection from policy $\pi$ is not possible, it is difficult to estimate $\rho_\pi(\boldsymbol{s})$ in Equation 1, and thus the expected value of the Q-function $\eta(\pi_\theta) := \int_{\mathcal{S}} \rho_\pi(\boldsymbol{s}) \int_{\mathcal{A}} \pi_\theta(\boldsymbol{a}|\boldsymbol{s}) Q^\pi(\boldsymbol{s}, \boldsymbol{a})$. Given a static

dataset $\mathcal{D}^\mu$ consisting of multiple trajectories $\{(\boldsymbol{s}_n, \boldsymbol{a}_n, r_n)\}$ collected by a behavior policy $\mu(\boldsymbol{a}|\boldsymbol{s})$, previous off-policy methods (Silver et al., 2014; Lillicrap et al., 2016) estimate $\eta(\pi_\theta)$ with a surrogate objective $\hat{\eta}(\pi_\theta)$ by replacing $\rho_\pi(\boldsymbol{s})$ with $\rho_\mu(\boldsymbol{s})$. In offline settings, due to the importance of sticking with the behavior policy, prior works (Peng et al., 2019; Nair et al., 2020) explicitly constrain the learned policy $\pi$ to be similar to $\mu$, while maximizing the expected value of the Q-functions:

$$\arg\max_\pi \quad \int_\mathcal{S} \rho_\mu(\boldsymbol{s}) \int_\mathcal{A} \pi(\boldsymbol{a}|\boldsymbol{s}) Q_\phi(\boldsymbol{s}, \boldsymbol{a}) \, \mathrm{d}\boldsymbol{a} \, \mathrm{d}\boldsymbol{s} - \frac{1}{\alpha} \int_\mathcal{S} \rho_\mu(\boldsymbol{s}) D_{\mathrm{KL}} \left( \pi(\cdot|\boldsymbol{s}) || \mu(\cdot|\boldsymbol{s}) \right) \mathrm{d}\boldsymbol{s}. \quad (2)$$

The first term in Equation 2 corresponds to the surrogate objective $\hat{\eta}(\pi_\theta)$, where $Q_\phi(\boldsymbol{s}, \boldsymbol{a})$ is a learned Q-function of the current policy $\pi$. The second term is a regularization term to constrain the learned policy within support of the dataset $\mathcal{D}^\mu$ with $\alpha$ being the coefficient.

## 2.2 POLICY IMPROVEMENT VIA WEIGHTED REGRESSION

The optimal policy $\pi^*$ for Equation 2 can be derived (Peters et al., 2010; Peng et al., 2019; Nair et al., 2020) by use of Lagrange multiplier:

$$\pi^*(\boldsymbol{a}|\boldsymbol{s}) = \frac{1}{Z(\boldsymbol{s})} \, \mu(\boldsymbol{a}|\boldsymbol{s}) \exp\left(\alpha Q_\phi(\boldsymbol{s}, \boldsymbol{a})\right), \quad (3)$$

where $Z(\boldsymbol{s})$ is the partition function. Equation 3 forms a policy improvement step.

Directly sampling from $\pi^*$ requires explicitly modeling behavior $\mu$, which itself is challenging in continuous action-space domains since $\mu$ can be very diverse. Prior methods (Peng et al., 2019; Wang et al., 2020; Chen et al., 2020) bypass this issue by projecting $\pi^*$ onto a parameterized policy $\pi_\theta$:

$$\arg\min_\theta \quad \mathbb{E}_{\boldsymbol{s}\sim\mathcal{D}^\mu} \left[ D_{\mathrm{KL}} \left( \pi^*(\cdot|\boldsymbol{s}) || \pi_\theta(\cdot|\boldsymbol{s}) \right) \right]$$

$$= \arg\max_\theta \quad \mathbb{E}_{(\boldsymbol{s},\boldsymbol{a})\sim\mathcal{D}^\mu} \left[ \frac{1}{Z(\boldsymbol{s})} \log \pi_\theta(\boldsymbol{a}|\boldsymbol{s}) \exp\left(\alpha Q_\phi(\boldsymbol{s}, \boldsymbol{a})\right) \right]. \quad (4)$$

Such a method is usually referred to as weighted regression, with $\exp\left(\alpha Q_\phi(\boldsymbol{s}, \boldsymbol{a})\right)$ being the regression weights.

Although weighted regression avoids the need to model the behavior policy explicitly, it requires calculating the exact density function $\pi_\theta(\boldsymbol{a}|\boldsymbol{s})$ as in Equation 4. This constrains the policy $\pi_\theta$ to distribution classes that have a tractable expression for the density function. We find this in practice limits the model expressivity and could be suboptimal in some cases (Section 3.1).

## 2.3 DIFFUSION PROBABILISTIC MODEL

Diffusion models (Sohl-Dickstein et al., 2015; Ho et al., 2020; Song et al., 2021b) are generative models by first defining a forward process to gradually add noise to an unknown data distribution $p_0(\boldsymbol{x}_0)$ and then learning to reverse it. The forward process $\{\boldsymbol{x}(t)\}_{t\in[0,T]}$ is defined by a stochastic differential equation (SDE) $\mathrm{d}\boldsymbol{x}_t = f(\boldsymbol{x}_t, t)\mathrm{d}t + g(t)\mathrm{d}\boldsymbol{w}_t$, where $\boldsymbol{w}_t$ is a standard Brownian motion and $f(t), g(t)$ are hand-crafted functions (Song et al., 2021b) such that the transition distribution $p_{t0}(\boldsymbol{x}_t|\boldsymbol{x}_0) = \mathcal{N}(\boldsymbol{x}_t|\alpha_t\boldsymbol{x}_0, \sigma_t^2\boldsymbol{I})$ for some $\alpha_t, \sigma_t > 0$ and $p_T(\boldsymbol{x}_T) \approx \mathcal{N}(\boldsymbol{x}_T|0, \boldsymbol{I})$. To reverse the forward process, diffusion models define a scored-based model $\boldsymbol{s}_\theta$ and optimize the parameter $\theta$ by:

$$\arg\min_\theta \quad \mathbb{E}_{t,\boldsymbol{x}_0,\boldsymbol{\epsilon}}[\|\sigma_t\boldsymbol{s}_\theta(\boldsymbol{x}_t, t) + \boldsymbol{\epsilon}\|_2^2], \quad (5)$$

where $t \sim \mathcal{U}(0, T)$, $\boldsymbol{x}_0 \sim p_0(\boldsymbol{x}_0)$, $\boldsymbol{\epsilon} \sim \mathcal{N}(0, \boldsymbol{I})$, $\boldsymbol{x}_t = \alpha_t\boldsymbol{x}_0 + \sigma_t\boldsymbol{\epsilon}$.

Sampling by diffusion models can be alternatively viewed as discretizing the diffusion ODEs (Song et al., 2021b), which are generally faster than discretizing the diffusion SDEs (Song et al., 2021a; Lu et al., 2022). Specifically, the sampling procedure needs to first sample a pure Gaussian $\boldsymbol{x}_T \sim \mathcal{N}(0, \boldsymbol{I})$, and then solve the following ODE from time $T$ to time $0$ by numerical ODE solvers:

$$\mathrm{d}\boldsymbol{x}_t = \left[ f(\boldsymbol{x}_t, t) - \frac{1}{2}g^2(t)\boldsymbol{s}_\theta(\boldsymbol{x}_t, t) \right] \mathrm{d}t. \quad (6)$$

Then the final solution $\boldsymbol{x}_0$ at time $0$ is the sample from the diffusion models.

## 3   METHOD

We propose a Selecting-from-Behavior-Candidates (SfBC) approach to address the limited expressivity problem in offline RL. Below we first motivate our method by highlighting the importance of a distributionally expressive policy in learning from diverse behaviors. Then we derive a high-level solution to this problem from a generative modeling perspective.

### 3.1   LEARNING FROM DIVERSE BEHAVIORS

In this section, we show that the weighted regression broadly used in previous works might limit the distributional expressivity of the policy and lead to performance degrade. As described in Section 2.2, conventional policy regression methods project the optimal policy $\pi^*$ in Equation 3 onto a parameterized policy set. In continuous action-space domains, the projected policy is usually limited to a narrow range of unimodal distributions (e.g., squashed Gaussian), whereas the behavior policy could be highly diverse (e.g., multimodal). Lack of expressivity directly prevents the RL agent from exactly mimicking a diverse behavior policy. This could eventually lead to sampling undesirable out-of-sample actions during policy evaluation and thus large extrapolation error. Even if Q-values can be accurately estimated, an inappropriate unimodal assumption about the optimal policy might still prevent extracting a policy that has multiple similarly rewarding but distinctive strategies.

We design a simple task named Bidirectional Car to better explain this point. Consider an environment where a car placed in the middle of two endpoints can go either side to gain the final reward. If an RL agent finds turning left and right similarly rewarding, by incorrectly assuming a unimodal distribution of the behavior policy, it ends up staying put instead of taking either one of the optimal actions (Figure 1). As a result, unimodal policies fail to completely solve this task or loss diversity whereas a more distributionally expressive policy easily succeeds.

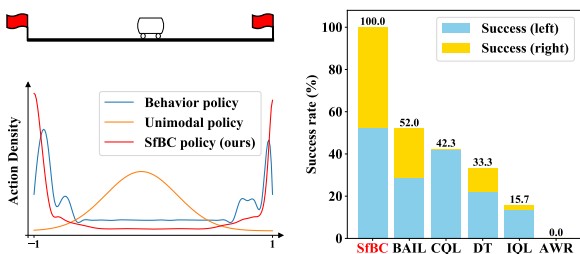

Figure 1: Illustration of the Bidirectional-Car task and comparison between SfBC and unimodal policies. See Section 6.2 for experimental details.

We therefore deduce that distributional expressivity is a necessity to enable diverse behavior learning. To better model the complex behavior policy, we need more powerful generative modeling for the policy distribution, instead of the simple and unimodal Gaussians.

### 3.2   SELECTING FROM BEHAVIOR CANDIDATES

In this section, we provide a generative view of how to model a potentially diverse policy. Specifically, in order to model $\pi^*$ with powerful generative models, essentially we need to perform maximum likelihood estimation for the model policy $\pi_\theta$, which is equivalent to minimizing KL divergence between the optimal and model policy:

$$\arg\max_\theta \quad \mathbb{E}_{\boldsymbol{s}\sim\mathcal{D}^\mu}\mathbb{E}_{a\sim\pi^*(\cdot|s)}\left[\log \pi_\theta(a|s)\right] \Leftrightarrow \arg\min_\theta \quad \mathbb{E}_{\boldsymbol{s}\sim\mathcal{D}^\mu}\left[D_{\mathrm{KL}}\left(\pi^*(\cdot|\boldsymbol{s})||\pi_\theta(\cdot|\boldsymbol{s})\right)\right]. \quad (7)$$

However, drawing samples directly from $\pi^*$ is difficult, so previous methods (Peng et al., 2019; Nair et al., 2020; Wang et al., 2020) rely on the weighted regression as described in Equation 4.

The main reason that limits the expressivity of $\pi_\theta$ is the need of calculating exact and derivable density function $\pi_\theta(\boldsymbol{a}|\boldsymbol{s})$ in policy regression, which places restrictions on distribution classes that we can choose from. Also, we might not know what the behavior or optimal policy looks like previously.

Our solution is based on a key observation that directly parameterizing the policy $\pi$ is not necessary. To better model a diverse policy, we propose to decouple the learning of $\pi$ into two parts. Specifically, we leverage Equation 3 to form a policy improvement step:

$$\pi(\boldsymbol{a}|\boldsymbol{s}) \propto \mu_\theta(\boldsymbol{a}|\boldsymbol{s}) \exp\left(\alpha Q_\phi(\boldsymbol{s}, \boldsymbol{a})\right). \quad (8)$$

One insight of the equation above is that minimizing KL divergence between $\mu$ and $\mu_\theta$ is much easier compared with directly learning $\pi_\theta$ because sampling from $\mu$ is straightforward given $D^\mu$. This

allows to us to leverage most existing advances in generative modeling (Section 4.1). $Q_\phi(s, a)$ could be learned using the existing Q-learning framework (Section 4.2).

The inverse temperature parameter $\alpha$ in Equation 8 serves as a trade-off between conservative and greedy improvement. We can see that when $\alpha \to 0$, the learned policy falls back to the behavior policy, and when $\alpha \to +\infty$ the learned policy becomes a greedy policy.

To sample actions from $\pi$, we use an importance sampling technique. Specifically, for any state $s$, first we draw $M$ action samples from a learned behavior policy $\mu_\theta(\cdot|s)$ as candidates. Then we evaluate these action candidates with a learned critic $Q_\phi$. Finally, an action is resampled from $M$ candidates with $\exp(\alpha Q_\phi(s, a))$ being the sampling weights. We summarize this procedure as selecting from behavior candidates (SfBC), which could be understood as an analogue to rejection sampling.

Although generative modeling of the behavior policy has been explored by several works (Fujimoto et al., 2019; Kumar et al., 2019), it was mostly used to form an explicit distributional constraint for the policy model $\pi_\theta$. In contrast, we show directly leveraging the learned behavior model to generate actions is not only feasible but beneficial on the premise that high-fidelity behavior modeling can be achieved. We give a practical implementation in the next section.

## 4 PRACTICAL IMPLEMENTATION

In this section, we derive a practical implementation of SfBC, which includes diffusion-based behavior modeling and planning-based Q-learning. An algorithm overview is given in Appendix A.

### 4.1 DIFFUSION-BASED BEHAVIOR MODELING

It is critical that the learned behavior model is of high fidelity because generating any out-of-sample actions would result in unwanted extrapolation error, while failing to cover all in-sample actions would restrict feasible action space for the policy. This requirement brings severe challenges to existing behavior modeling methods, which mainly include using Gaussians or VAEs. Gaussian models suffer from limited expressivity as we have discussed in Section 3.1. VAEs, on the other hand, need to introduce a variational posterior distribution to optimize the model distribution, which has a trade-off between the expressivity and the tractability (Kingma et al., 2016; Lucas et al., 2019). This still limits the expressivity of the model distribution. An empirical study is given in Section 6.3.

To address this problem, we propose to learn from diverse behaviors using diffusion models (Ho et al., 2020), which have recently achieved great success in modeling diverse image distributions (Ramesh et al., 2022; Saharia et al., 2022), outperforming other generative models (Dhariwal & Nichol, 2021). Specifically, we follow Song et al. (2021b) and learn a state-conditioned diffusion model $s_\theta$ to predict the time-dependent noise added to the action $a$ sampled from the behavior policy $\mu(\cdot|s)$:

$$\theta = \arg\min_\theta \quad \mathbb{E}_{(s,a) \sim D^\mu, \epsilon, t}[\|\sigma_t s_\theta(\alpha_t a + \sigma_t \epsilon, s, t) + \epsilon\|_2^2], \tag{9}$$

where $\epsilon \sim \mathcal{N}(0, I)$, $t \sim \mathcal{U}(0, T)$. $\alpha_t$ and $\sigma_t$ are determined by the forward diffusion process. Intuitively $s_\theta$ is trained to denoise $a_t := \alpha_t a + \sigma_t \epsilon$ into the unperturbed action $a$ such that $a_T \sim \mathcal{N}(0, I)$ can be transformed into $a \sim \mu_\theta(\cdot|s)$ by solving an inverse ODE defined by $s_\theta$ (Equation 6).

### 4.2 Q-LEARNING VIA IN-SAMPLE PLANNING

Generally, Q-learning can be achieved via the Bellman expectation operator:

$$\mathcal{T}^\pi Q(s, a) = r(s, a) + \gamma \mathbb{E}_{s' \sim P(\cdot|s,a), a' \sim \pi(\cdot|s')} Q(s', a'). \tag{10}$$

However, $\mathcal{T}^\pi$ is based on one-step bootstrapping, which has two drawbacks: First, this can be computationally inefficient due to its dependence on many steps of extrapolation. This drawback is exacerbated in diffusion settings since drawing actions from policy $\pi$ in Equation 10 is also time-consuming because of many iterations of Langevin-type sampling. Second, estimation errors may accumulate over long horizons. To address these problems, we take inspiration from episodic learning methods (Blundell et al., 2016; Ma et al., 2022) and propose a planning-based operator $\mathcal{T}_\mu^\pi$:

$$\mathcal{T}_\mu^\pi Q(s, a) := \max_{n \geq 0} \{(\mathcal{T}^\mu)^n \mathcal{T}^\pi Q(s, a)\}, \tag{11}$$

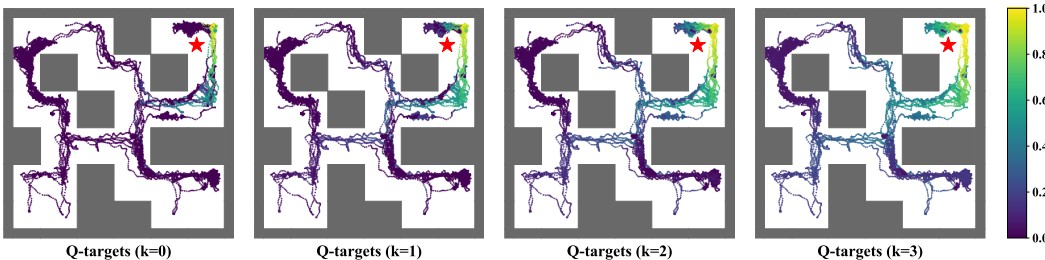

Figure 2: Visualizations of the implicitly planned Q-targets $R_n^{(k)}$ sampled from the dataset of an AntMaze task in four consecutive value iterations. The red pentagram stands for the reward signal. Implicit planning helps to iteratively stitch together successful subtrajectories.

where $\mu$ is the behavior policy. $\mathcal{T}_\mu^\pi$ combines the strengths of both the n-step operator $(\mathcal{T}^\mu)^n$, which enjoys a fast contraction property, and the operator $\mathcal{T}^\pi$, which has a more desirable fixed point. We prove in Appendix C that $\mathcal{T}_\mu^\pi$ is also convergent, and its fixed point is bounded between $Q^\pi$ and $Q^*$.

Practically, given a dataset $\mathcal{D}^\mu = \{(s_n, a_n, r_n)\}$ collected by behavior $\mu$, with $n$ being the timestep in a trajectory. We can rewrite Equation 11 in a recursive manner to calculate the Q-learning targets:

$$R_n^{(k)} = r_n + \gamma \max(R_{n+1}^{(k)}, V_{n+1}^{(k-1)}), \tag{12}$$

$$\text{where} \quad V_n^{(k-1)} := \mathbb{E}_{\boldsymbol{a} \sim \pi(\cdot|\boldsymbol{s}_n)} Q_\phi(\boldsymbol{s}_n, \boldsymbol{a}), \tag{13}$$

$$\text{and} \quad \phi = \arg\min_\phi \quad \mathbb{E}_{(\boldsymbol{s}_n, \boldsymbol{a}_n) \sim \mathcal{D}^\mu} \|Q_\phi(\boldsymbol{s}_n, \boldsymbol{a}_n) - R_n^{(k-1)}\|_2^2. \tag{14}$$

Above $k \in \{1, 2, \dots\}$ is the iteration number. We define $R_n^{(0)}$ as the vanilla return of trajectories. Equation 12 offers an implicit planning scheme within dataset trajectories that mainly helps to avoid bootstrapping over unseen actions and to accelerate convergence. Equation 13 enables the generalization of actions in similar states across different trajectories (stitching together subtrajectories). Note that we have omitted writing the iteration superscript of $\pi$ and $\mu$ for simplicity. During training, we alternate between calculating new Q-targets $R_n$ and fitting the action evaluation model $Q_\phi$.

Although the operator $\mathcal{T}_\mu^\pi$ is inspired by the multi-step estimation operator $\mathcal{T}_{\text{vem}}$ proposed by Ma et al. (2022). They have notable differences in theoretical properties. First, $\mathcal{T}_{\text{vem}}$ can only apply to deterministic environments, while our method also applies to stochastic settings. Second, unlike $\mathcal{T}_{\text{vem}}$, $\mathcal{T}_\mu^\pi$ does not share the same fixed point with $\mathcal{T}^\pi$. We compare two methods in detail in Appendix G.

## 5 RELATED WORK

**Reducing extrapolation error in offline RL**. Offline RL typically requires careful trade-offs between maximizing expected returns and staying close to the behavior policy. Once the learned policy deviates from the behavior policy, extrapolation error will be introduced in dynamic programming, leading to performance degrade (Fujimoto et al., 2019). Several works propose to address this issue by introducing either policy regularization on the distributional discrepancy with the behavior policy (Fujimoto et al., 2019; Kumar et al., 2019; Wu et al., 2019; Fujimoto & Gu, 2021), or value pessimism about unseen actions (Kumar et al., 2020; Kostrikov et al., 2021). Another line of research directly extracts policy from the dataset through weighted regression, hoping to avoid selecting unseen actions (Peng et al., 2019; Nair et al., 2020; Wang et al., 2020). However, some recent works observe that the trade-off techniques described above are not sufficient to reduce extrapolation error, and propose to learn Q-functions through expectile regression without ever querying policy-generated actions (Kostrikov et al., 2022; Ma et al., 2022). Unlike them, We find that limited policy expressivity is the main reason that introduces extrapolation error in previous weighted regression methods, and use an expressive policy model to help reduce extrapolation error.

**Dynamic programming over long horizons**. Simply extracting policies from behavior Q-functions can yield good performance in many D4RL tasks because it avoids dynamic programming and therefore the accompanied extrapolation error (Peng et al., 2019; Chen et al., 2020; Brandfonbrener

| Dataset | Environment | SfBC (Ours) | IQL | VEM | AWR | BAIL | BCQ | CQL | DT | Diffuser |
|---|---|---|---|---|---|---|---|---|---|---|
| Medium-Expert | HalfCheetah | **92.6 ± 0.5** | 86.7 | - | 52.7 | 72.2 | 64.7 | 62.4 | 86.8 | 79.8 |
| Medium-Expert | Hopper | **108.6 ± 2.1** | 91.5 | - | 27.1 | **106.2** | 100.9 | 98.7 | **107.6** | **107.2** |
| Medium-Expert | Walker | **109.8 ± 0.2** | **109.6** | - | 53.8 | **107.2** | 57.5 | **111.0** | **108.1** | **108.4** |
| Medium | HalfCheetah | 45.9 ± 2.2 | **47.4** | **47.4** | 37.4 | 30.0 | 40.7 | 44.4 | 42.6 | 44.2 |
| Medium | Hopper | 57.1 ± 4.1 | **66.3** | 56.6 | 35.9 | 62.2 | 54.5 | 58.0 | **67.6** | 58.5 |
| Medium | Walker | 77.9 ± 2.5 | **78.3** | 74.0 | 17.4 | 73.4 | 53.1 | **79.2** | 74.0 | **79.7** |
| Medium-Replay | HalfCheetah | 37.1 ± 1.7 | **44.2** | - | 40.3 | 40.3 | 38.2 | **46.2** | 36.6 | 42.2 |
| Medium-Replay | Hopper | 86.2 ± 9.1 | **94.7** | - | 28.4 | **94.7** | 33.1 | 48.6 | 82.7 | **96.8** |
| Medium-Replay | Walker | 65.1 ± 5.6 | **73.9** | - | 15.5 | 58.8 | 15.0 | 26.7 | 66.6 | 61.2 |
| Average (Locomotion) | | 75.6 | 76.9 | - | 34.3 | 71.6 | 51.9 | 63.9 | 74.7 | 75.3 |
| Default | AntMaze-umaze | **92.0 ± 2.1** | 87.5 | 87.5 | 56.0 | 85.0 | 78.9 | 74.0 | 59.2 | - |
| Diverse | AntMaze-umaze | **85.3 ± 3.6** | 62.2 | 78.0 | 70.3 | 76.7 | 55.0 | **84.0** | 53.0 | - |
| Play | AntMaze-medium | **81.3 ± 2.6** | 71.2 | 78.0 | 0.0 | 15.0 | 0.0 | 61.2 | 0.0 | - |
| Diverse | AntMaze-medium | **82.0 ± 3.1** | 70.0 | 77.0 | 0.0 | 23.3 | 0.0 | 53.7 | 0.0 | - |
| Play | AntMaze-large | **59.3 ± 14.3** | 39.6 | 57.0 | 0.0 | 0.0 | 6.7 | 15.8 | 0.0 | - |
| Diverse | AntMaze-large | 45.5 ± 6.6 | 47.5 | **58.0** | 0.0 | 8.3 | 2.2 | 14.9 | 0.0 | - |
| Average (AntMaze) | | **74.2** | 63.0 | **72.6** | 21.0 | 46.7 | 23.8 | 50.6 | 18.7 | - |
| Average (Maze2d) | | 74.0 | 50.0 | - | 10.8 | - | 9.1 | 7.7 | - | **119.5** |
| Average (FrankaKitchen) | | **57.1** | 53.3 | - | 8.7 | - | 11.7 | 48.2 | - | - |
| Both-side | Bidirectional-Car | **100.0 ± 0.0** | 15.7 | 0.0 | 0.0 | 52.0 | 88.0 | 42.3 | 33.3 | - |
| Single-side | Bidirectional-Car | **100.0 ± 0.0** | 100.0 | 100.0 | 96.3 | 100.0 | 100.0 | 100.0 | 100.0 | - |

Table 1: Evaluation numbers of SfBC. Scores are normalized according to Fu et al. (2020). Numbers within 5 percent of the maximum in every individual task are highlighted in boldface. Experiment and evaluation details are provided in Appendix B. We report scores with 15 diffusion steps.

et al., 2021). However, Kostrikov et al. (2022) shows this method performs poorly in tasks that require stitching together successful subtrajectories (e.g., Maze-like environments). Such tasks are also challenging for methods based on one-step bootstrapping because they might require hundreds of steps to reach the reward signal, with the reward discounted and estimation error accumulated along the way. Episodic memory-based methods address this problem by storing labeled experience in the dataset, and plans strictly within the trajectory to update evaluations of every decision (Blundell et al., 2016; Hu et al., 2021; Ma et al., 2022). The in-sample planning scheme allows dynamic programming over long horizons to suppress the accumulation of extrapolation error, which inspires our method.

**Generative models for behavior modeling**. Cloning diverse behaviors in a continuous action space requires powerful generative models. In offline RL, several works (Fujimoto et al., 2019; Kumar et al., 2019; Wu et al., 2019; Zhou et al., 2021; Chen et al., 2022) have tried using generative models such as Gaussians or VAEs to model the behavior policy. However, the learned behavior model only serves as an explicit distributional constraint for another policy during training. In broader RL research, generative adversarial networks (Goodfellow et al., 2020), masked autoencoders (Germain et al., 2015), normalizing flows (Dinh et al., 2016), and energy-based models (Du & Mordatch, 2019) have also been used for behavior modeling (Ho & Ermon, 2016; Ghasemipour et al., 2021; Singh et al., 2020; Liu et al., 2020). Recently, diffusion models (Ho et al., 2020) have achieved great success in generating diverse and high-fidelity image samples (Dhariwal & Nichol, 2021). However, exploration of its application in behavior modeling is still limited. Janner et al. (2022) proposes to solve offline tasks by iteratively denoising trajectories, while our method uses diffusion models for single-step decision-making. Concurrently with our work, Wang et al. (2022) also studies applying diffusion models to offline RL to improve policy expressivity. However, they use diffusion modeling as an implicit regularization during training of the desired policy instead of an explicit policy prior.

## 6 EXPERIMENTS

### 6.1 EVALUATIONS ON D4RL BENCHMARKS

In Table 1, we compare the performance of SfBC to multiple offline RL methods in several D4RL (Fu et al., 2020) tasks. `MuJoCo locomotion` is a classic benchmark where policy-generated datasets only cover a narrow part of the state-action space, so avoiding querying out-of-sample actions is

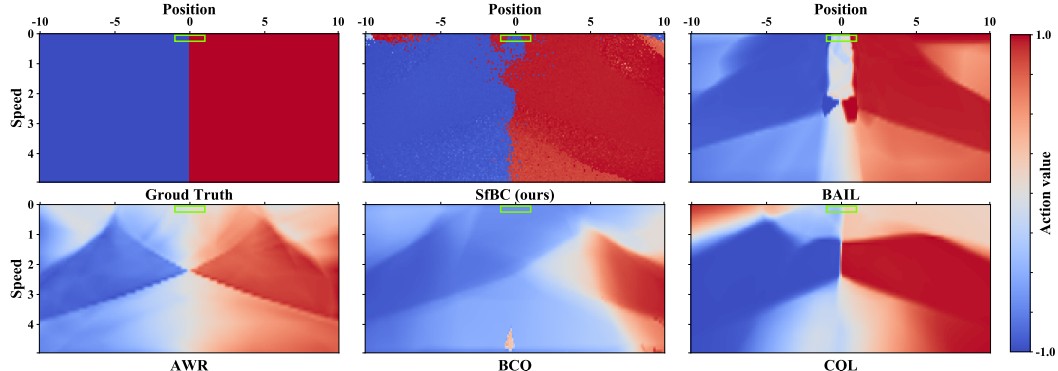

Figure 3: Visualizations of actions taken by different RL agents in the Bidirectional-Car task. The ground truth corresponds to an agent which always takes the best actions, which is either 1.0 or -1.0. White space indicates suboptimal decisions. Green bounding boxes indicate possible initial states.

critical (Fujimoto et al., 2019; Kumar et al., 2020). The Medium dataset of this benchmark is generated by a single agent, while the Medium-Expert and the Medium-Replay dataset are generated by a mixture of policies. `AntMaze` is about an ant robot navigating itself in a maze, which requires both low-level robot control and high-level navigation. Since the datasets consist of undirected trajectories, solving AntMaze typically requires the algorithm to have strong "stitching" ability (Fu et al., 2020). Different environments contain mazes of different sizes, reflecting different complexity. `Maze2d` is very similar to AntMaze except that it's about a ball navigating in a maze instead of an ant robot. `FrankaKitchen` are robot-arm manipulation tasks. We only focus on the analysis of MuJoCo locomotion and AntMaze tasks due to the page limit. Our choices of referenced baselines are detailed in Appendix E.

Overall, SfBC outperforms most existing methods by large margins in complex tasks with sparse rewards such as AntMaze. We notice that VEM also achieves good results in AntMaze tasks and both methods share an implicit in-sample planning scheme, indicating that episodic planning is effective in improving algorithms' stitching ability and thus beneficial in Maze-like environments. In easier locomotion tasks, SfBC provides highly competitive results compared with state-of-the-art algorithms. It can be clearly shown that performance gain is large in datasets generated by a mixture of distinctive policies (Medium-Expert) and is relatively small in datasets that are highly uniform (Medium). This is reasonable because SfBC is motivated to better model diverse behaviors.

## 6.2 LEARNING FROM DIVERSE BEHAVIORS

In this section, we analyze the benefit of modeling behavior policy using highly expressive generative models. Although SfBC outperforms baselines in many D4RL tasks. The improvement is mainly incremental, but not decisive. We attribute this to the lack of multiple optimal solutions in existing benchmarks. To better demonstrate the necessity of introducing an expressive generative model, we design a simple task where a heterogeneous dataset is collected in an environment that allows two distinctive optimal policies.

**Bidirectional-Car task**. As depicted in Figure 1, we consider an environment where a car is placed in the middle of two endpoints. The car chooses an action in the range [-1,1] at each step, representing throttle, to influence the direction and speed of the car. The speed of the car will *monotonically* increase based on the absolute value of throttle. The direction of the car is determined by the sign of the current throttle. Equal reward will be given on the arrival of either endpoint within the rated time. It can be inferred with ease that, in any state, the optimal decision should be either 1 or -1, which is not a unimodal distribution. The collected dataset also contains highly diverse behaviors, with an approximately equal number of trajectories ending at both endpoints. For the comparative study, we collect another dataset called "Single-Side" where the only difference from the original one is that we remove all trajectories ending at the left endpoint from the dataset.

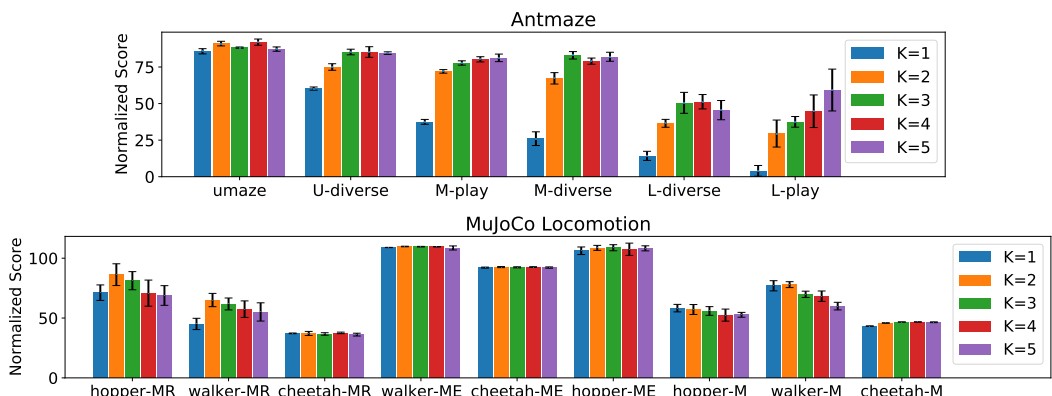

Figure 4: Ablation studies of the value iteration number $K$ in MuJoCo Locomotion and Antmaze domains. $K = 1$ represents algorithms that use vanilla returns $R_n^{(0)}$ as Q-learning targets without the implicit planning technique. All results are averaged over 4 independent random seeds.

We test our method against several baselines, with the results given in Table 1. Among all referenced methods, SfBC is the only one that can always arrive at either endpoint within rated time in the Bidirectional-Car environment, whereas most methods successfully solve the "Single-Side" task. To gain some insight into why this happens, we illustrate the decisions made by an SfBC agent and other RL agents in the 2-dimensional state space. As is shown in Figure 3, the SfBC agent selects actions of high absolute values at nearly all states, while other unimodal actors fail to pick either one of the optimal actions when presented with two distinctive high-rewarding options. Therefore, we conclude that an expressive policy is necessary for performing diverse behavior learning.

### 6.3 ABLATION STUDIES

**Diffusion vs. other generative models**. Our first ablation study aims to evaluate 3 variants of SfBC which are respectively based on diffusion models (Ho et al., 2020), Gaussian probabilistic models, and latent-based models (VAEs, Kingma & Welling (2014)). The three variants use exactly the same training framework with the only difference being the behavior modeling method. As is shown in Table 4 in Appendix D, the diffusion-based policy outperforms the other two variants by a clear margin in most experiments, especially in tasks with heterogeneous datasets (e.g., Medium-Expert), indicating that diffusion models are fit for "high-fidelity" behavior modeling.

**Implicit in-sample planning**. To study the importance of implicit in-sample planning on the performance of SfBC, we first visualize the estimated state values learned at different iterations of Q-learning in an AntMaze environment (Figure 2). We can see that implicit planning helps to iteratively stitch together successful subtrajectories and provides optimistic action evaluations. Then we aim to study how the value iteration number $K$ affects the performance of the algorithm in various environments. As shown in Figure 4, we compare the performance of $K$ in the range $\{1, 2, 3, 4, 5\}$ and find that implicit planning is beneficial in complex tasks like AntMaze-Medium and AntMaze-Large. However, it is less important in MuJoCo-locomotion tasks. This finding is consistent with a prior work (Brandfonbrener et al., 2021).

## 7 CONCLUSION

In this work, we address the problem of limited policy expressivity in previous weighted regression methods by decoupling the policy model into a behavior model and an action evaluation model. Such decoupling allows us to use a highly expressive diffusion model for high-fidelity behavior modeling, which is further combined with a planning-based operator to reduce extrapolation error. Our method enables learning from a heterogeneous dataset in a continuous action space while avoiding selecting out-of-sample actions. Experimental results on the D4RL benchmark show that our approach outperforms state-of-the-art algorithms in most tasks. With this work, we hope to draw attention to the application of high-capacity generative models in offline RL.

## Reproducibility

To ensure that our work is reproducible, we submit the source code as supplementary material. We also provide the pseudo-code of our algorithm in Appendix A and implementation details of our algorithm in Appendix B.

## Acknowledgement

We thank Shiyu Huang, Yichi Zhou, and Hao Hu for discussing. This work was supported by the National Key Research and Development Program of China (2020AAA0106000, 2020AAA0106302, 2021YFB2701000), NSFC Projects (Nos. 62061136001, 62076147, U19B2034, U1811461, U19A2081, 61972224), BNRist (BNR2022RC01006), Tsinghua Institute for Guo Qiang, and the High Performance Computing Center, Tsinghua University.

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

## A ALGORITHM OVERVIEW

---

**Algorithm 1** Selecting from Behavior Candidates (Training)

---

Initialize the score-based model $s_\theta$, the action evaluation model $Q_\phi$

Calculate vanilla discounted returns $R_n^{(0)}$ for every state-action pair in dataset $\mathcal{D}^\mu$

*// Training the behavior model*

**for** each gradient step **do**

  Sample $B$ data points $(\boldsymbol{s}, \boldsymbol{a})$ from $\mathcal{D}^\mu$, $B$ Gaussian noises $\epsilon$ from $\mathcal{N}(0, \boldsymbol{I})$ and $B$ time $t$ from $\mathcal{U}(0, T)$

  Perturb $\boldsymbol{a}$ according to $\boldsymbol{a}_t := \alpha_t \boldsymbol{a} + \sigma_t \boldsymbol{\epsilon}$

  Update $\theta \leftarrow \lambda_s \nabla_\theta \sum[\|\sigma_t \mathbf{s}_\theta(\boldsymbol{a}_t, \boldsymbol{s}, t) + \boldsymbol{\epsilon}\|_2^2]$

**end for**

*// Training the action evaluation model iteratively*

**for** iteration $k = 1$ **to** $K$ **do**

  Initialize training parameters $\phi$ of the action evaluation model $Q_\phi$

  **for** each gradient step **do**

    Sample $B$ data points $\left(\boldsymbol{s}, \boldsymbol{a}, R^{(k-1)}\right)$ from $\mathcal{D}^\mu$

    Update $\phi \leftarrow \phi - \lambda_Q \nabla_\phi \sum[\|Q_\phi(\boldsymbol{s}, \boldsymbol{a}) - R^{(k-1)}\|_2^2]$

  **end for**

  *// Update the Q-training targets as in Algorithm 2*

  $R^{(k)} = \text{Planning}(\mathcal{D}^\mu, \mu_\theta, Q_\phi)$

**end for**

---

**Algorithm 2** Implicit In-sample Planning

---

Input a behavior dataset $\mathcal{D}^\mu$ (sequentially ordered), a learned behavior policy $\mu_\theta$, a critic model $Q_\phi$

*// Evaluate every state in dataset according to Equation 13 with M Monte Carlo samples (parallelized)*

**for** each minibatch $\{\boldsymbol{s}_n\}$ splitted from $\mathcal{D}^\mu$ **do**

  Sample M actions $\hat{\boldsymbol{a}}_n^{1:M}$ from $\mu_\theta(\cdot|\boldsymbol{s}_n)$, and calculate Q-values $\hat{R}_n^{1:M} = Q_\phi(\boldsymbol{s}_n, \hat{\boldsymbol{a}}_n^{1:M})$

  Calculate state value $V_n = \sum_m \left[\exp\left(\alpha \hat{R}_n^m\right) \hat{R}_n^m\right] / \sum_m \exp\left(\alpha \hat{R}_n^m\right)$

**end for**

*// Performing implicit in-sample planning recursively*

**for** timestep $n = \|\mathcal{D}^\mu\|$ **to** 0 **do**

  $R_n = r_n + \gamma \max(R_{n+1}, V_{n+1})$ if $n$ is not the last episode step, else $r_n$

**end for**

Output the new Q-training targets $\{R_n\}$

---

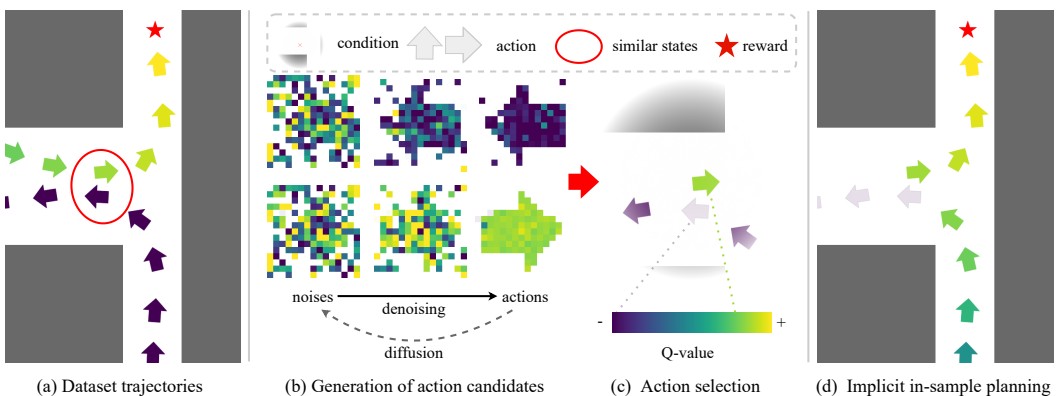

(a) Dataset trajectories   (b) Generation of action candidates   (c) Action selection   (d) Implicit in-sample planning

Figure 5: An algorithm overview of SfBC.

# B  EXPERIMENTAL DETAILS

## B.1  IMPLEMENTATION DETAILS OF SFBC

**Network Architecture.** SfBC includes a conditional scored-based model which estimates the score function of the behavior action distribution, and an action evaluation model which outputs the Q-values of given state-action pairs. The architecture of the behavior model resembles U-Nets, but with spatial convolutions changed to simple dense connections, inspired by Janner et al. (2022). For the action evaluation model, we use a 2-layer MLP with 256 hidden units and SiLU activation functions. The same network architecture is applied across all tasks except for AntMaze-Large, where we add an extra layer of 512 hidden units for the action evaluation model.

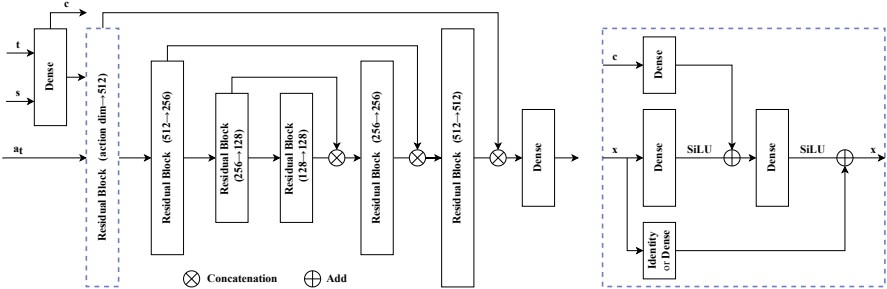

Figure 6: The network architecture of the behavior model.

**Behavior training.** In all experiments, we use the Adam optimizer and a batch size of 4096. The conditional scored-based model is trained for 500 data epochs with a learning rate of 1e-4. For the data perturbation method, we use Variance Preserving (VP) SDE as introduced in Song et al. (2021b), where we have $\mathrm{d}\boldsymbol{x} = -\frac{1}{2}\beta(t)\boldsymbol{x}\mathrm{d}t + \sqrt{\beta(t)}\mathrm{d}\mathbf{w}$, such that we have $f(\boldsymbol{x}, t) = -\frac{1}{2}\beta(t)\boldsymbol{x}$ and $g(t) = \sqrt{\beta(t)}$ in Equation 6, and also:

$$p_{t0}(\boldsymbol{x}_t|\boldsymbol{x}_0) = \mathcal{N}(\boldsymbol{x}_t|\alpha_t\boldsymbol{x}_0, \sigma_t^2\boldsymbol{I}) = \mathcal{N}(\boldsymbol{x}_t|e^{-\frac{1}{2}\int_0^t \beta(s)\mathrm{d}s}\boldsymbol{x}_0, [1 - e^{-\int_0^t \beta(s)\mathrm{d}s}]\boldsymbol{I}). \tag{15}$$

Following the default settings in Song et al. (2021b), we set $\beta(t) = (\beta_{\max} - \beta_{\min})\, t + \beta_{\min}$, with $\beta_{\min}$ being 0.1 and $\beta_{\max}$ being 20.

**Action evaluation via in-sample planning.** The action evaluation model is trained for 100 data epochs with a learning rate of 1e-3 for each value iteration. We use $K = 2$ value iterations for all MuJoCo tasks, $K = 4$ for Antmaze-umaze tasks, and $K = 5$ for other Antmaze tasks. In each iteration, new Q-targets will be recalculated according to Equation 12 and Equation 13 based on the latest policy. We use Monte Carlo methods and importance sampling to estimate $V_n^{(k-1)}$ in Equation 13:

$$
\begin{aligned}
V_n^{(k-1)} &= \mathbb{E}_{\boldsymbol{a}\sim\pi(\cdot|\boldsymbol{s}_n)} Q_\phi(\boldsymbol{s}_n, \boldsymbol{a}) \\
&= \mathbb{E}_{\boldsymbol{a}\sim\mu_\theta(\cdot|\boldsymbol{s}_n)} \frac{\exp\left(\alpha Q_\phi(\boldsymbol{s}_n, \boldsymbol{a})\right)}{Z(\boldsymbol{s}_n)} Q_\phi(\boldsymbol{s}_n, \boldsymbol{a}) \\
&\approx \sum_M \left[\frac{\exp\left(\alpha Q_\phi(\boldsymbol{s}_n, \boldsymbol{a})\right)}{\sum_M \exp\left(\alpha Q_\phi(\boldsymbol{s}_n, \boldsymbol{a})\right)} Q_\phi(\boldsymbol{s}_n, \boldsymbol{a})\right],
\end{aligned} \tag{16}
$$

with the inverse temperature $\alpha$ set to 20 and the Monte Carlo sample number set to 16 in all tasks. Note that at the beginning of each value iteration, we normalize Q-targets stored in the dataset and reinitialize the training parameters of the action evaluation model. Different from most prior works (Fujimoto et al., 2019; Kumar et al., 2020; Ma et al., 2022; Kostrikov et al., 2022), we do not use either ensembled networks or target networks to stabilize Q-learning.

**Diffusion sampling.** To draw action samples from the behavior model, we use a 3rd-order specialized diffusion ODE solver proposed by Lu et al. (2022) to solve the inverse ODE problem in Equation 6. We use a diffusion step of $D = 15$ for all reported results in Table 1, which is significantly less than

the typical 35-50 diffusion steps required if using ordinary **RK45** ODE solver (Dormand & Prince, 1980). We also compare the performance and runtime of diffusion steps in the range $\{5, 10, 15, 25\}$ with the results reported in Table 2. Generally, we find that 10-25 diffusion steps perform similarly well in MuJoco Locomotion tasks and 15-25 diffusion steps perform similarly well in Antmaze tasks.

**Evaluation.** Following the evaluation metric proposed by Fu et al. (2020), we run all Antmaze and MuJoCo experiments over 4 trials (different random seeds) and other experiments over 3 trials. For each trial, performance is averaged on another 100 test seeds for Antmaze tasks and 20 test seeds for other tasks at regular intervals (5 data epochs). During algorithm evaluation, we select actions in a deterministic way. Specifically, the action with the highest Q-value within $M$ behavior candidates will be selected for environment inference during evaluation. In MuJoCo Locomotion tasks, we average four actions with the highest Q-values among all candidates and find this technique helps to stabilize performance. We set the candidate number $M$ to 32 in all experiments.

**Runtime.** We test the runtime of our algorithm on a RTX 2080Ti GPU. For algorithm training, the runtime cost of training the behavior model is 10.5 hours for 600 epochs, and the runtime cost of training the action evaluation model is about 31 minutes for each value iteration, (usually 2-5 iterations, 1M data points considered). For a concrete example, it roughly takes 155 minutes to train the action evaluation model (K=5) and 10.5 hours to train the behavior model for the "halfcheetah-medium" task.

As for the evaluation runtime, theoretically, SfBC requires at least $D$ times of network inference time compared with non-diffusion methods ($D = 1$), $D$ being the diffusion steps. To accelerate algorithm evaluation, we implement a parallel evaluation scheme similar to Clemente et al. (2017); Weng et al. (2022) that could allow evaluating the algorithm under multiple test seeds at the same time, allowing us to significantly reduce the evaluation runtime by utilizing the parallel computing power of GPUs (Figure 7).

| Diffusion Steps $D$ | 5 steps | 10 steps | 15 steps | 25 steps |
|---|---|---|---|---|
| Performance (Locomotion) | 2.3 | 72.9 | 75.6 | 74.4 |
| Performance (Antmaze) | 5.5 | 65.7 | 74.2 | 73.0 |
| Runtime (1 episode, # envs=1) | 22.3 s | 38.0 s | 50.0 s | 93.0 s |
| Runtime (1 episode, # envs=20) | 1.5 s | 2.5 s | **3.2 s** | 5.0 s |

Table 2: Ablation studies of the diffusion steps. The runtime is reported for the "halfcheetah-medium" task on a RTX 2080Ti GPU. 1 episode stands for 1000 environment steps.

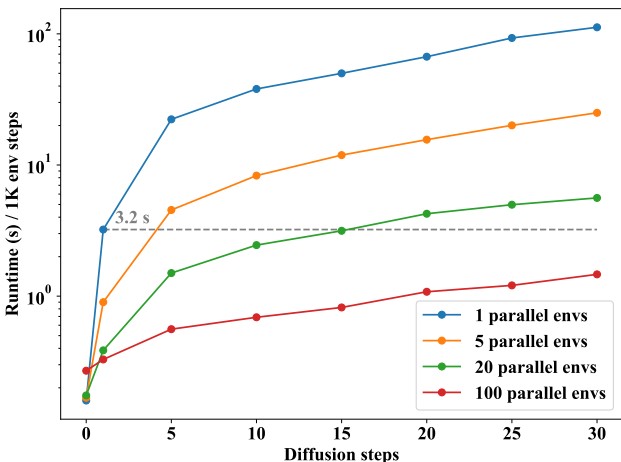

Figure 7: Evaluation runtime of SfBC.

### B.2 IMPLEMENTATION DETAILS FOR ABLATION STUDIES

**SfBC + VAEs/Gaussians**. For the VAE-based behavior model, we use exactly the same network architecture and training loss as Fujimoto et al. (2019) and train the behavior model for 300k iterations at a learning rate of 3e-4. For the Gaussian-based policy model, we follow Nair et al. (2020) and use a 4-layer MLP with 256 hidden units and ReLU activation functions. The sampled action from the parameterized Gaussian distribution is squashed to range $[-1, 1]$ by a Tanh activation function. The Gaussian behavior model is trained by directly maximizing the log-likelihood of the dataset distribution for 300k iterations at a learning rate of 3e-4. Other experiment settings are consistent with the diffusion-based method.

**SfBC - Planning**. Removing the planning-based procedure from SfBC is equivalent to performing SfBC for only one iteration, which only learns a behavior Q-function purely from vanilla returns. Other than this, we use the same network architecture and training paradigm as SfBC.

### B.3 SOURCES OF REFERENCED BASELINE NUMBERS

For IQL (Kostrikov et al., 2022), D4RL performance numbers are reported in its original paper, except for Maze2d tasks, which we reference Janner et al. (2022). The performance in the Bidirectional-Car task is based on a PyTorch reimplementation of the algorithm (`https://github.com/gwthomas/IQL-PyTorch`). We use the same hyperparameters as in the original paper for MuJoCo Locomotion tasks.

For VEM (Ma et al., 2022) and Diffuser (Janner et al., 2022), all D4RL performance numbers come from their respective papers. Performance numbers of VEM in the Bidirectional-Car task are based on a slightly modified version of the algorithm's official codebase (`https://github.com/YiqinYang/VEM`). Since the performance of VEM is very sensitive to a hyperparameter $\tau$ in their algorithm. We evaluate $\tau \in \{0.1, 0.2, ..., 0.9\}$ and report the best-performing choice.

For AWR (Peng et al., 2019), BCQ (Fujimoto et al., 2019) and CQL (Kumar et al., 2020), all their D4RL performance numbers come from Fu et al. (2020). Their performance numbers in the Bidirectional-Car task are based on three independent codebases: `https://github.com/Farama-Foundation/D4RL-Evaluations` for AWR, `https://github.com/sfujim/BCQ` for BCQ and `https://github.com/young-geng/CQL` for CQL. We mostly use the default hyperparameters in their respective codebases.

For BAIL (Chen et al., 2020), all reported performance numbers come from our experiments based on a slightly modified version of its official codebase (`https://github.com/lanyavik/BAIL`). Note that BAIL proposes a technique to replace oracle returns with augmented returns in MuJoCo Locomotion tasks, whereas we omit using this technique because it cannot be easily applied to other offline tasks. Other than this, we use default settings in the original codebase.

For DT (Chen et al., 2021), D4RL performance numbers are reported in DT's paper, except for AntMaze tasks, which we reference Kostrikov et al. (2022). The performance numbers in the Bidirectional-Car task are based on the algorithm's official codebase (`https://github.com/kzl/decision-transformer`). We use the same hyperparameters as they did for MuJoCo Locomotion tasks.

## C  THEORETICAL ANALYSIS

In this section, we provide some theoretical analysis of our planning-based operator:

$$\mathcal{T}_\mu^\pi Q(\boldsymbol{s}, \boldsymbol{a}) := \max_{n \geq 0}\{(\mathcal{T}^\mu)^n \mathcal{T}^\pi Q(\boldsymbol{s}, \boldsymbol{a})\}. \tag{17}$$

First, we provide the following proposition to discuss the contraction property of $\mathcal{T}_\mu^\pi$ and the bound of its fixed point.

**Proposition 1.** *We have the following properties of $\mathcal{T}_\mu^\pi$.*

*1) $\mathcal{T}_\mu^\pi$ owns monotonicity, i.e., for $\forall Q_1 \leq Q_2$, we have $\mathcal{T}_\mu^\pi Q_1 \leq \mathcal{T}_\mu^\pi Q_2$.*

*2) $\mathcal{T}_\mu^\pi$ is at least a $\gamma$-contraction.*

*3) Assume the fixed point of $\mathcal{T}_\mu^\pi$ is $\tilde{Q}$, then we have $Q^\pi(\boldsymbol{s}, \boldsymbol{a}) \leq \tilde{Q}(\boldsymbol{s}, \boldsymbol{a}) \leq Q^*(\boldsymbol{s}, \boldsymbol{a})$ holds for $\forall \boldsymbol{s}, \boldsymbol{a}$, here $Q^\pi, Q^*$ are the fixed points of $\mathcal{T}^*$ and $\mathcal{T}^\pi$ respectively.*

*Proof.*
1) For $\forall Q_1 \leq Q_2, \forall \boldsymbol{s}, \boldsymbol{a}, \forall n \in \mathbb{N}$, we have

$$(\mathcal{T}^\mu)^n \mathcal{T}^\pi Q_1(\boldsymbol{s}, \boldsymbol{a}) \leq (\mathcal{T}^\mu)^n \mathcal{T}^\pi Q_2(\boldsymbol{s}, \boldsymbol{a}). \quad \text{(by monotonicity of } \mathcal{T}^\pi \text{ and } (\mathcal{T}^\mu)^n) \tag{18}$$

Thus we have

$$\mathcal{T}_\mu^\pi Q_1(\boldsymbol{s}, \boldsymbol{a}) = \max_{n \geq 0}\{(\mathcal{T}^\mu)^n \mathcal{T}^\pi Q_1(\boldsymbol{s}, \boldsymbol{a})\} \leq \max_{n \geq 0}\{(\mathcal{T}^\mu)^n \mathcal{T}^\pi Q_2(\boldsymbol{s}, \boldsymbol{a})\} = \mathcal{T}_\mu^\pi Q_2(\boldsymbol{s}, \boldsymbol{a}). \tag{19}$$

2) For $\forall Q_1, Q_2, \forall \boldsymbol{s}, \boldsymbol{a}$, we have

$$\begin{aligned}
\left|\mathcal{T}_\mu^\pi Q_1(\boldsymbol{s}, \boldsymbol{a}) - \mathcal{T}_\mu^\pi Q_2(\boldsymbol{s}, \boldsymbol{a})\right| &= \left|\max_{n \geq 0}\{(\mathcal{T}^\mu)^n \mathcal{T}^\pi Q_1(\boldsymbol{s}, \boldsymbol{a})\} - \max_{n \geq 0}\{(\mathcal{T}^\mu)^n \mathcal{T}^\pi Q_2(\boldsymbol{s}, \boldsymbol{a})\}\right| \\
&\leq \max_{n \geq 0}\{|(\mathcal{T}^\mu)^n \mathcal{T}^\pi Q_1(\boldsymbol{s}, \boldsymbol{a}) - (\mathcal{T}^\mu)^n \mathcal{T}^\pi Q_2(\boldsymbol{s}, \boldsymbol{a})|\} \\
&\leq \max_{n \geq 0}\{\gamma^{n+1} \|Q_1 - Q_2\|_\infty\} \\
&= \gamma \|Q_1 - Q_2\|_\infty.
\end{aligned} \tag{20}$$

Consequently, $\mathcal{T}_\mu^\pi$ is at least a $\gamma$-contraction.

3) For $\forall \boldsymbol{s}, \boldsymbol{a}$, we first prove $Q^\pi(\boldsymbol{s}, \boldsymbol{a}) \leq \tilde{Q}(\boldsymbol{s}, \boldsymbol{a})$. For $\forall m \in \mathbb{N}$, we have

$$\begin{aligned}
Q^\pi(\boldsymbol{s}, \boldsymbol{a}) = \mathcal{T}^\pi Q^\pi(\boldsymbol{s}, \boldsymbol{a}) &\leq \mathcal{T}_\mu^\pi Q^\pi(\boldsymbol{s}, \boldsymbol{a}) = \mathcal{T}_\mu^\pi \mathcal{T}^\pi Q^\pi(\boldsymbol{s}, \boldsymbol{a}) \\
&\leq \mathcal{T}_\mu^\pi \mathcal{T}_\mu^\pi Q^\pi(\boldsymbol{s}, \boldsymbol{a}) \quad \text{(by monotonicity of } \mathcal{T}_\mu^\pi) \\
&\leq ... \leq (\mathcal{T}_\mu^\pi)^m Q^\pi(\boldsymbol{s}, \boldsymbol{a}), \\
\text{Thus} \quad Q^\pi(\boldsymbol{s}, \boldsymbol{a}) &\leq \lim_{m \to \infty} (\mathcal{T}_\mu^\pi)^m Q^\pi(\boldsymbol{s}, \boldsymbol{a}) = \tilde{Q}(\boldsymbol{s}, \boldsymbol{a}).
\end{aligned} \tag{21}$$

Now we prove that $\tilde{Q}(\boldsymbol{s}, \boldsymbol{a}) \leq Q^*(\boldsymbol{s}, \boldsymbol{a})$. We have

$$\begin{aligned}
\mathcal{T}^\pi Q^*(\boldsymbol{s}, \boldsymbol{a}) &= \mathcal{R}(\boldsymbol{s}, \boldsymbol{a}) + \gamma \mathbb{E}_{s'} \mathbb{E}_{a' \sim \hat{\pi}(\cdot|s')} Q^*(\boldsymbol{s}', \boldsymbol{a}') \\
&\leq \mathcal{R}(\boldsymbol{s}, \boldsymbol{a}) + \gamma \mathbb{E}_{s'} \max_{a'} Q^*(\boldsymbol{s}', \boldsymbol{a}') = Q^*(\boldsymbol{s}, \boldsymbol{a}).
\end{aligned} \tag{22}$$

Similarly, we have $\mathcal{T}^\mu Q^*(\boldsymbol{s}, \boldsymbol{a}) \leq Q^*(\boldsymbol{s}, \boldsymbol{a})$.

Then for $\forall n, m \in \mathbb{N}$,

$$\begin{aligned}
Q^*(\boldsymbol{s}, \boldsymbol{a}) \geq \mathcal{T}^\mu Q^*(\boldsymbol{s}, \boldsymbol{a}) &\geq (\mathcal{T}^\mu)^2 Q^*(\boldsymbol{s}, \boldsymbol{a}) \quad \text{(by monotonicity of } \mathcal{T}^\mu) \\
&\geq ... \geq (\mathcal{T}^\mu)^n Q^*(\boldsymbol{s}, \boldsymbol{a}) \\
&\geq (\mathcal{T}^\mu)^n \mathcal{T}^\pi Q^*(\boldsymbol{s}, \boldsymbol{a}), \quad \text{(by monotonicity of } (\mathcal{T}^\mu)^n) \\
\text{Thus} \quad Q^*(\boldsymbol{s}, \boldsymbol{a}) &\geq \mathcal{T}_\mu^\pi Q^*(\boldsymbol{s}, \boldsymbol{a}) \\
&\geq \mathcal{T}_\mu^\pi \mathcal{T}_\mu^\pi Q^*(\boldsymbol{s}, \boldsymbol{a}) \quad \text{(by monotonicity of } \mathcal{T}_\mu^\pi) \\
&\geq ... \geq (\mathcal{T}_\mu^\pi)^m Q^*(\boldsymbol{s}, \boldsymbol{a}), \\
\text{Thus} \quad Q^*(\boldsymbol{s}, \boldsymbol{a}) &\geq \lim_{m \to \infty} (\mathcal{T}_\mu^\pi)^m Q^*(\boldsymbol{s}, \boldsymbol{a}) = \tilde{Q}(\boldsymbol{s}, \boldsymbol{a}).
\end{aligned} \tag{23}$$

$\square$

Moreover, similar to the analysis in Ma et al. (2022), we provide the following proposition to show that at the beginning of the training when the current Q function estimates $Q(s, a)$ is significantly pessimistic, our $\mathcal{T}_\mu^\pi$ provides a relatively optimistic update and can contract the estimation error more quickly.

**Proposition 2.** *In practice, we consider* $\mathcal{T}_\mu^\pi Q(s, a) := \max_{0 \leq n \leq N}\{(\mathcal{T}^\mu)^n \mathcal{T}^\pi Q(s, a)\}$. *Then we have:*

$$|\mathcal{T}_\mu^\pi Q(s, a) - Q^*(s, a)| \leq \gamma^{n^*(s, a)}\|Q - \tilde{Q}_{n^*}\|_\infty + \|\tilde{Q}_{n^*} - Q^*\|_\infty, \quad \forall s, a, \qquad (24)$$

*here* $n^*(s, a) = \arg\max_{0 \leq n \leq N}\{(\mathcal{T}^\mu)^n \mathcal{T}^\pi Q(s, a)\}$ *and* $\tilde{Q}_{n^*}$ *is the fixed point of* $(\mathcal{T}^\mu)^{n^*(s, a)} \mathcal{T}^\pi$.

*Proof.* We can use the triangle inequality to prove this result

$$\begin{aligned}
&|\mathcal{T}_\mu^\pi Q(s, a) - Q^*(s, a)|\\
=&|(\mathcal{T}^\mu)^{n^*(s, a)} \mathcal{T}^\pi Q(s, a) - Q^*(s, a)|\\
\leq&|(\mathcal{T}^\mu)^{n^*(s, a)} \mathcal{T}^\pi Q(s, a) - (\mathcal{T}^\mu)^{n^*(s, a)} \mathcal{T}^\pi \tilde{Q}_{n^*}(s, a)| + |(\mathcal{T}^\mu)^{n^*(s, a)} \mathcal{T}^\pi \tilde{Q}_{n^*}(s, a) - Q^*(s, a)|\\
=&|(\mathcal{T}^\mu)^{n^*(s, a)} \mathcal{T}^\pi Q(s, a) - (\mathcal{T}^\mu)^{n^*(s, a)} \mathcal{T}^\pi \tilde{Q}_{n^*}(s, a)| + |\tilde{Q}_{n^*}(s, a) - Q^*(s, a)|\\
\leq&\gamma^{n^*(s, a)}\|Q - \tilde{Q}_{n^*}\|_\infty + \|\tilde{Q}_{n^*} - Q^*\|_\infty.
\end{aligned}$$

$$(25)$$

$\square$

When $Q$ is significantly lower than $\tilde{Q}_{n^*}, Q^*$, $\|\tilde{Q}_{n^*} - Q^*\|_\infty$ is often conspicuously lower than $\|Q - \tilde{Q}_{n^*}\|_\infty$ and $n^*(s, a)$ is relatively large (this often happens at the beginning of the training since the initial Q estimates are often near zero and thus pessimistic). At this time, based on this proposition, our operator $\mathcal{T}_\mu^\pi$, could contract the estimation error with a rate of around $\gamma^{n^*(s, a)}$, which could significantly reduce extrapolation iterations required.

# D    MISSING PERFORMANCE NUMBERS

| Dataset | Environment | SfBC (Ours) | IQL | VEM | AWR | BAIL | BCQ | CQL | DT | Diffuser |
|---------|-------------|-------------|-----|-----|-----|------|-----|-----|----|---------|
| Sparse | Maze2d-umaze | $73.9 \pm 6.6$ | 47.4 | - | 1.0 | - | 12.8 | 5.7 | - | **113.9** |
| Sparse | Maze2d-medium | $73.8 \pm 2.9$ | 34.9 | - | 7.6 | - | 8.3 | 5.0 | - | **121.5** |
| Sparse | Maze2d-large | $74.4 \pm 1.7$ | 58.6 | - | 23.7 | - | 6.2 | 12.5 | - | **123.0** |
| **Average (Maze2d)** | | 74.0 | 50.0 | - | 10.8 | - | 9.1 | 7.7 | - | 119.5 |
| Complete | FrankaKitchen | $\mathbf{77.9 \pm 0.6}$ | 62.5 | - | 0.0 | - | 8.11 | 43.8 | - | - |
| Partial | FrankaKitchen | $\mathbf{47.9 \pm 4.1}$ | **46.3** | - | 15.4 | - | 18.9 | **49.8** | - | - |
| Mixed | FrankaKitchen | $45.4 \pm 1.6$ | **51.0** | - | 10.6 | - | 8.1 | **51.0** | - | - |
| **Average (FrankaKitchen)** | | **57.1** | 53.3 | - | 8.7 | - | 11.7 | 48.2 | - | - |

Table 3: Additional performance numbers of SfBC in Maze2d and FrankaKitchen tasks. We report the mean and standard deviation over three seeds for SfBC. Scores are normalized according to Fu et al. (2020).

| Dataset | Environment | SfBC | SfBC + Gaussian | SfBC + VAE | SfBC - Planning |
|---------|-------------|------|-----------------|------------|-----------------|
| Medium-Expert | HalfCheetah | $\mathbf{92.6 \pm 0.5}$ | $79.4 \pm 1.4$ | $85.2 \pm 2.9$ | $\mathbf{91.4 \pm 0.6}$ |
| Medium-Expert | Hopper | $\mathbf{108.6 \pm 2.1}$ | $\mathbf{107.8 \pm 7.8}$ | $92.0 \pm 7.3$ | $\mathbf{109.0 \pm 1.0}$ |
| Medium-Expert | Walker | $\mathbf{109.8 \pm 0.2}$ | $71.5 \pm 1.5$ | $\mathbf{109.3 \pm 2.5}$ | $\mathbf{109.4 \pm 0.9}$ |
| Medium | HalfCheetah | $\mathbf{45.9 \pm 2.2}$ | $42.0 \pm 0.2$ | $43.4 \pm 0.1$ | $42.4 \pm 0.2$ |
| Medium | Hopper | $57.1 \pm 4.1$ | $58.1 \pm 1.5$ | $\mathbf{65.6 \pm 3.3}$ | $60.1 \pm 4.2$ |
| Medium | Walker | $77.9 \pm 2.5$ | $\mathbf{82.4 \pm 1.1}$ | $79.1 \pm 2.5$ | $80.3 \pm 0.9$ |
| Medium-Replay | HalfCheetah | $37.1 \pm 1.7$ | $36.2 \pm 1.2$ | $\mathbf{42.4 \pm 0.5}$ | $37.5 \pm 0.6$ |
| Medium-Replay | Hopper | $\mathbf{86.2 \pm 9.1}$ | $67.8 \pm 6.5$ | $58.6 \pm 4.8$ | $58.6 \pm 1.3$ |
| Medium-Replay | Walker | $\mathbf{65.1 \pm 5.6}$ | $65.8 \pm 4.4$ | $62.2 \pm 4.3$ | $62.6 \pm 2.2$ |
| **Average** | | **75.6** | 67.9 | 70.9 | **72.3** |
| Default | AntMaze-umaze | $92.0 \pm 2.1$ | $93.3 \pm 2.4$ | $91.6 \pm 2.4$ | $\mathbf{96.7 \pm 4.7}$ |
| Diverse | AntMaze-umaze | $85.3 \pm 3.6$ | $\mathbf{88.3 \pm 2.4}$ | $78.3 \pm 4.7$ | $80.0 \pm 10.8$ |
| Play | AntMaze-medium | $\mathbf{81.3 \pm 2.6}$ | $80.0 \pm 4.1$ | $68.3 \pm 2.4$ | $35.0 \pm 4.1$ |
| Diverse | AntMaze-medium | $82.0 \pm 3.1$ | $85.0 \pm 7.1$ | $65.0 \pm 7.1$ | $33.3 \pm 6.2$ |
| Play | AntMaze-large | $\mathbf{59.3 \pm 14.3}$ | $43.3 \pm 7.1$ | $35.0 \pm 8.2$ | $8.3 \pm 8.5$ |
| Diverse | AntMaze-large | $\mathbf{45.5 \pm 6.6}$ | $26.7 \pm 8.5$ | $20.0 \pm 0.0$ | $6.7 \pm 4.7$ |
| **Average** | | **74.2** | 69.4 | 59.7 | 43.3 |

Table 4: Ablations of generative modeling methods and the implicit planning method. We report the mean and standard deviation over four seeds for the main experiment and three seeds for other experiments. Scores are normalized according to Fu et al. (2020).

# E    CHOICES OF REFERENCED BASELINES

Referenced baselines methods of SfBC can be roughly divided into four categories: 1. Policy regression methods that require dynamic programming such as IQL (Kostrikov et al., 2022) and VEM (Ma et al., 2022). 2. Policy regression methods that use vanilla returns as regression weights such as AWR (Peng et al., 2019) and BAIL (Chen et al., 2020). 3. Adaptations of existing off-policy algorithms with policy regularization such as BCQ (Fujimoto et al., 2019) and CQL (Kumar et al., 2020). 4. Sequence modeling methods such as DT (Chen et al., 2021) and Diffuser (Janner et al., 2022). Here we further highlight several methods which bear some resemblance to our approach: Both IQL and SfBC aim to entirely avoid selecting out-of-sample actions, except that IQL uses weighted regression while SfBC does not; VEM also uses an implicit in-sample planning scheme similar to ours; BCQ also uses a generative model (VAE) for behavior modeling, but only to assist the learning of another policy model; Diffuser, like SfBC, is also a diffusion-based algorithm, but uses approximated guided sampling at trajectory level instead of importance sampling at step level.

# F  TRAINING CURVES

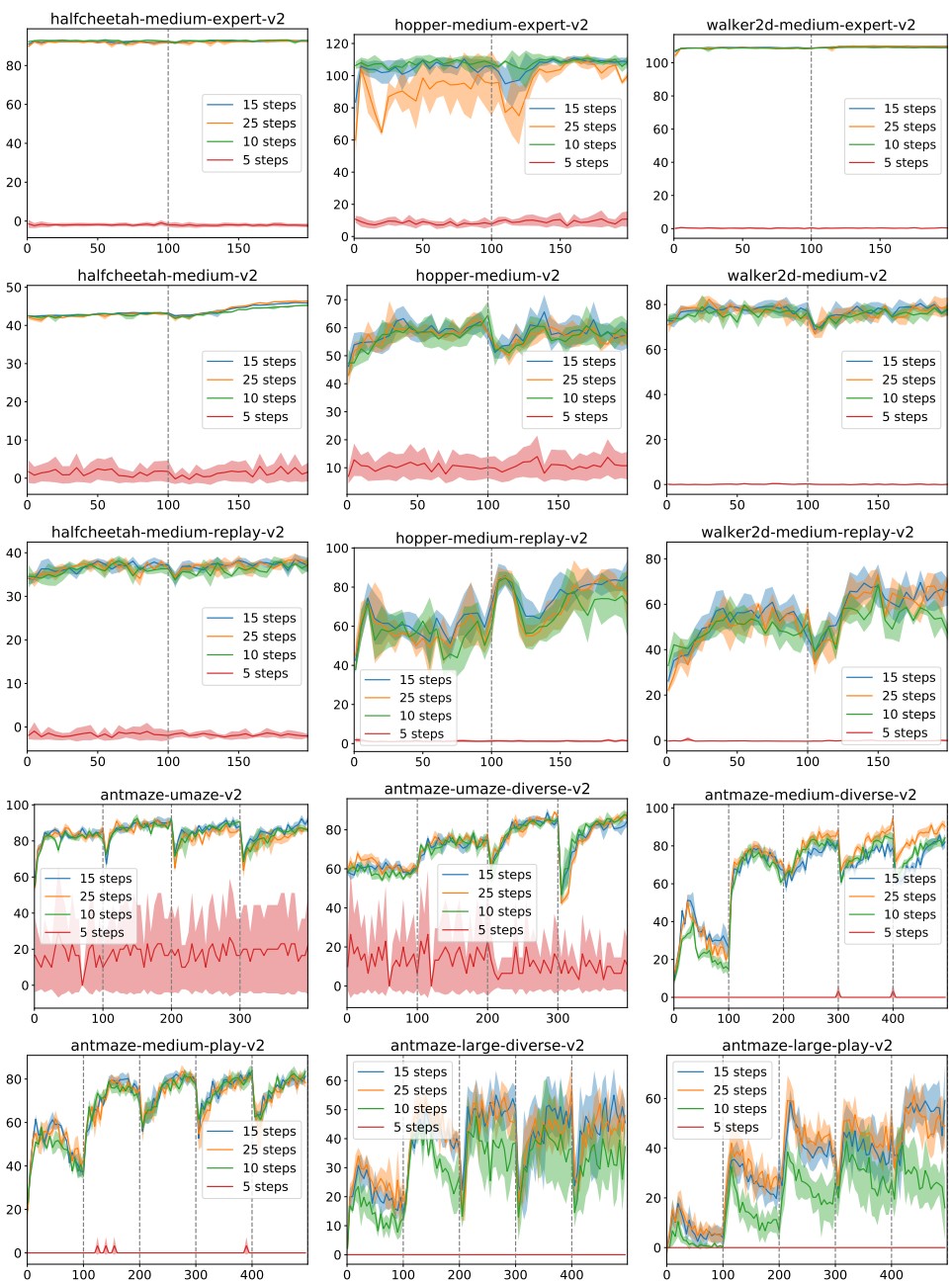

Figure 8: Training curves of SfBC for MuJoCo and Antmaze tasks with different diffusion steps. We report the mean and standard deviation over four seeds for all experiments. Note that the parameters of the critic model are initialized at the beginning of each value iteration (every 100 data epochs).

## G CONNECTIONS TO PRIOR WORKS

In this section, we discuss in more detail the connections between SfBC and two prior works, namely VEM (Ma et al., 2022) and EMAQ (Ghasemipour et al., 2021).

### G.1 VEM

Our in-sample planning-based Q-operator $\mathcal{T}_\mu^\pi$ bears some similarity to the multi-step estimation operator $\mathcal{T}_{\text{vem}}$ proposed by Ma et al. (2022). A simplified version of $\mathcal{T}_{\text{vem}}$ is defined as:

$$\mathcal{T}_{\text{vem}}V(\boldsymbol{s}) := \max_{n \geq 0}\{(\mathcal{T}^\mu)^n \mathcal{T}_\mu^\tau V(\boldsymbol{s})\}, \tag{26}$$

which is is built on $\mathcal{T}_\mu^\tau$, an expectile-based V-learning operator proposed by VEM:

$$\mathcal{T}_\mu^\tau V(\boldsymbol{s}) := \mathbb{E}_{\boldsymbol{a} \sim \mu(\cdot|\boldsymbol{s})} \left\{ \begin{array}{ll} \tau[r(\boldsymbol{s},\boldsymbol{a}) + \gamma V(\boldsymbol{s}')] + (1-\tau)V(\boldsymbol{s}) & \text{if} \quad r(\boldsymbol{s},\boldsymbol{a}) + \gamma V(\boldsymbol{s}') \geq V(\boldsymbol{s}) \\ (1-\tau)[r(\boldsymbol{s},\boldsymbol{a}) + \gamma V(\boldsymbol{s}')] + \tau V(\boldsymbol{s}) & \text{if} \quad r(\boldsymbol{s},\boldsymbol{a}) + \gamma V(\boldsymbol{s}') < V(\boldsymbol{s}) \end{array} \right\} \tag{27}$$

here $\tau \in [0,1)$ is a hyperparameter that helps interpolate the Bellman expectation operator $\mathcal{T}^\mu$ ($\tau = 0.5$) and the Bellman optimality operator $\mathcal{T}^*$ ($\tau \to 1.0$). $V(\cdot)$ is an arbitrary scalar function. $\mathcal{T}_\mu^\tau$ has some nice properties such as monotonicity ($\mathcal{T}_\mu^{\tau_2}V(\boldsymbol{s}) > \mathcal{T}_\mu^{\tau_1}V(\boldsymbol{s})$ always holds for any $V$ given $\tau_2 > \tau_1$). With these properties, Ma et al. (2022) derives that $\mathcal{T}_{\text{vem}}$ and $\mathcal{T}_\mu^\tau$ share the same fixed point.

However, VEM cannot be applied to stochastic environments because Equation 27 requires comparing $V(\boldsymbol{s})$ and $r(\boldsymbol{s},\boldsymbol{a}) + \gamma V(\boldsymbol{s}')$. While $V(\boldsymbol{s})$ is a scalar given $\boldsymbol{s}$, $r(\boldsymbol{s},\boldsymbol{a}) + \gamma V(\boldsymbol{s}')$ is a random variable since $\boldsymbol{s}' \sim P(\cdot|\boldsymbol{s},\boldsymbol{a})$. To fix this problem, VEM simply assumes that the environment is deterministic, namely $r(\boldsymbol{s},\boldsymbol{a})$ and $P(\cdot|\boldsymbol{s},\boldsymbol{a})$ are all Dirac.

Compared with VEM, our in-sample planning-based Q-operator $\mathcal{T}_\mu^\pi$ is not dependent on the expectile-based V-operator $\mathcal{T}_\mu^\tau$, but uses an hypothetically improved policy $\pi > \mu$ for optimistic planning:

$$\mathcal{T}_\mu^\pi Q(\boldsymbol{s},\boldsymbol{a}) := \max_{n \geq 0}\{(\mathcal{T}^\mu)^n \mathcal{T}^\pi Q(\boldsymbol{s},\boldsymbol{a})\}, \tag{28}$$

which does not require the environment to be deterministic. A disadvantage of using $\mathcal{T}^\pi$ to replace $\mathcal{T}_\mu^\tau$ is that we no longer have the monotonicity property (e.g., $\mathcal{T}^\pi Q(\boldsymbol{s},\boldsymbol{a}) > \mathcal{T}^\mu Q(\boldsymbol{s},\boldsymbol{a})$ always holds for any $Q$). However, we can still derive that the fixed point of $\mathcal{T}_\mu^\pi$ is bounded between $Q^\pi$ and $Q^*$ (See Appendix C for detailed results and proofs).

### G.2 EMAQ

The high-level idea of the selecting-from-behavior-candidates approach bears some resemblance to the Expected-Max Q-Learning (EMaQ) algorithm proposed by (Ghasemipour et al., 2021). EMaQ is built upon BCQ (Fujimoto et al., 2019), which computes the training target in Q-Learning by:

$$\mathcal{T}_{\text{BCQ}}^* Q(\boldsymbol{s},\boldsymbol{a}) := r(\boldsymbol{s},\boldsymbol{a}) + \gamma \max_{\boldsymbol{a}' \sim \mu_\theta(\cdot|\boldsymbol{s}')}\{Q(\boldsymbol{s}',\boldsymbol{a}' + \xi_\phi(\boldsymbol{s}',\boldsymbol{a}'))\}, \tag{29}$$

where $\xi_\phi(\boldsymbol{s},\boldsymbol{a})$ is an explicitly constrained perturbation network that helps relax the constraint of behavior policy $\mu$. The core motivation for EMaQ is to remove the perturbation model $\xi_\phi(\boldsymbol{s},\boldsymbol{a})$, by taking max over N Q-function evaluations:

$$\mathcal{T}_{\text{EMaQ}}Q(\boldsymbol{s},\boldsymbol{a}) := r(\boldsymbol{s},\boldsymbol{a}) + \gamma \mathbb{E}_{\{\boldsymbol{a}_i'\}^N \sim \mu_\theta(\cdot|\boldsymbol{s}')}[\max_{\boldsymbol{a}' \in \{\boldsymbol{a}_i'\}^N} Q(\boldsymbol{s}',\boldsymbol{a}_i')]. \tag{30}$$

For EMaQ, $N$ serves as a hyperparameter to interpolate $\mathcal{T}^\mu$ and $\mathcal{T}^*$. When $N = 1$, $\mathcal{T}_{\text{EMaQ}}$ becomes $\mathcal{T}^\mu$. When $N \to \infty$, $\mathcal{T}_{\text{EMaQ}}$ approaches $\mathcal{T}^*$ because $\{\boldsymbol{a}_i'\}^N$ nearly covers the whole action space.

In contrast, for SfBC, the hyperparameter $N$ is the number of Monte Carlo samples used to estimate the training Q-targets:

$$\begin{aligned} \mathcal{T}^\pi Q(\boldsymbol{s},\boldsymbol{a}) &= r(s,a) + \gamma \mathbb{E}_{\boldsymbol{a}' \sim \pi(\cdot|\boldsymbol{s}')}Q(\boldsymbol{s}',\boldsymbol{a}') \\ &= r(s,a) + \gamma \mathbb{E}_{\boldsymbol{a}' \sim \mu(\cdot|\boldsymbol{s}')}\frac{\exp{(\alpha Q(\boldsymbol{s}',\boldsymbol{a}'))}}{Z(\boldsymbol{s}')}Q(\boldsymbol{s}',\boldsymbol{a}') \\ &\approx r(s,a) + \gamma \sum_N \left[\frac{\exp{(\alpha Q(\boldsymbol{s}',\boldsymbol{a}'))}}{\sum_N \exp{(\alpha Q(\boldsymbol{s}',\boldsymbol{a}'))}}Q(\boldsymbol{s}',\boldsymbol{a}')\right] \end{aligned}$$

