# OpenReview forum: "Offline Reinforcement Learning via High-Fidelity Generative Behavior Modeling"
_ICLR.cc/2023/Conference — ICLR 2023 poster_

### Official Review · Reviewer_dfM5 · 2022-10-23

**Confidence:** 4
**Correctness:** 3
**Technical Novelty And Significance:** 2
**Empirical Novelty And Significance:** 2
**Recommendation:** 6

**Clarity, Quality, Novelty And Reproducibility:**

The paper needs to be improve in terms of clarify. Some important prior work is missing. Given a limited discussion of implementation details, it might be difficult to reproduce the results in the paper.

Please see the strengths and weakness for more details

**Strength And Weaknesses:**

# Strength
* The approach is simple and produces strong results on the standard benchmark.

# Weaknesses
* Right now, the main weakness is a lack of clarity. It is not clear how the method is implemented, and the paper can be significantly improved by expanding the implementation details.
* The approach has a limited novelty. Expressive policies have been previously studied in EMAQ [1]. Moreover, the value computation technique is similar to a soft version of EMAQ.
* The related work section misses some important prior work in particular [1], which also introduces a similar approach.
* At the end of section 4.2, it's mentioned that the n-step operator proposed in this work works in the stochastic setting. However, this statement is not completely clear. For example, does it mean that it's unbiased?


[1] EMaQ: Expected-Max Q-Learning Operator for Simple Yet Effective Offline and Online RL
Seyed Kamyar Seyed Ghasemipour, Dale Schuurmans, Shixiang Shane Gu

**Summary Of The Paper:**

The paper introduces a method for offline reinforcement learning that involves fitting a more expressive density model and an episodic planning technique. The method demonstrates decent results on D4RL, a standard benchmark for offline reinforcement learning.

**Summary Of The Review:**

In overall it's a simple a well-performing method. However, it has limited novelty and the writing needs to be improved before the paper can be accepted.

---

> ### Author Response · Authors · 2022-11-14
> **Response (Part 1/3): A review of EMaQ**
>
> We thank the reviewer for the very constructive comments by mentioning a very important related work EMaQ[2], which we missed during the literature-search phase. Before we give a point-to-point response, we think it would be helpful to first present a brief review of EMaQ. A more detailed (also more nicely formatted) review and comparison are presented in Appendix G.2 in the updated manuscript.
>
> **A brief review of EMaQ.**
>
> EMaQ is built upon BCQ, which computes the training target in Q-Learning by:
>
> $
>     \mathcal{T}^* _{BCQ} Q(s, a) := r(s, a) + \gamma \max _{a' \sim \mu_\theta(\cdot|s')}\{Q(s', a' + \xi_\phi(s', a'))\},
> $
>
> where $\xi_\phi(s, a)$ is an explicitly constrained perturbation network that helps relax the constraint of behavior policy $\mu_\theta$.
>
> The core motivation for EMaQ is to remove the perturbation model $\xi_\phi(s, a)$, by taking max over N Q-function evaluations:
> $
>     \mathcal{T}_{EMaQ} Q(s, a) := r(s, a) + \gamma \mathbb{E} _{ [a_i'] ^N \sim  {\mu}_\theta (\cdot | s')  }[\max _{a_i ' \in [a_i ']_N} Q(s', a_i ')].
> $
>
> For EMaQ, $N$ serves as a hyperparameter to interpolate $\mathcal{T} ^\mu$ and $\mathcal{T}^*$. When $N=1$, $\mathcal{T} _{EMaQ}$ becomes $\mathcal{T} ^\mu$. When $N \rightarrow \infty$, $\mathcal{T} _{EMaQ}$ approaches $\mathcal{T} ^*$ because $[a_i']^N$ nearly covers the whole action space.
>
> In contrast, for SfBC, the hyperparameter $N$ is the number of Monte Carlo samples used to estimate the training Q-targets:
>
> $
>     \mathcal{T}^\pi_{SfBC} Q(s, a) \approx r(s, a) + \gamma \sum_N \bigg[\frac{\mathrm{exp}\left(\alpha Q(s',a') \right)}{\sum_N \mathrm{exp}\left(\alpha Q(s',a') \right)}Q(s',a')\bigg]
> $
>
> Although $\mathcal{T}^\pi$ may look like a softmax version of EMaQ, which is only briefly mentioned but not studied in detail in Appendix J in [1], **it is not an interpolation of** $\mathcal{T}^\mu$ **and** $\mathcal{T}^*$. Changing $N$ only influence the variance of the Monte Carlo estimate for $\mathcal{T}^\pi Q(s, a)$, but does not affect the fixed point of $\mathcal{T}^\pi$.
>
> Besides, SfBC uses a multi-step planning-based operator $\mathcal{T}_\mu^\pi Q(s,a) := \max _{n \geq 0}\{(\mathcal{T}^{\mu})^{n}\mathcal{T}^{\pi}Q(s, a)\}$, while EMaQ only considers  one-step bootstrapping.
>
> During inference, both methods select actions from candidates generated by a behavior model. However, the focus and the claimed contribution of EMaQ and SfBC are almost entirely different. EMaQ focuses on studying how the choices of hyperparameter $N$ affect the performance of the algorithm and only mentions that they empirically find an expressive policy model yield good performance when explaining why they choose an autoregressive masked autoencoder architecture (MADE) over VAE in Section (Section 3.5). In contrast, SfBC makes much more effort to discuss how offline RL may benefit from a much more expressive behavior policy model.
>
> [2] EMaQ: Expected-Max Q-Learning Operator for Simple Yet Effective Offline and Online RL. https://arxiv.org/abs/2007.11091

---

> ### Author Response · Authors · 2022-11-14
> **Response (Part 2/3)**
>
> **Q1: The related work section misses some important prior work in particular ...**
>
> **A1:** We appreciate the reviewer for this very insightful comment. EMaQ is indeed a very important related work, which we have missed during the literature-search phase. We now have mentioned EMaQ in the Introduction Section, Method Section, and Related works Section in our updated manuscript. We have also added a new section in the Appendix to discuss in more detail about the connection and difference between the two works.
>
> **Q2: The approach has a limited novelty. Expressive policies have been previously studied in EMAQ.**
>
> **A2:** EMaQ and our research both find that offline RL could benefit greatly from an expressive behavior policy. However, we believe it would be more accurate to state that expressive policies have been discussed or mentioned (instead of studied) in EMaQ. **SfBC and EMaQ have easily distinguishable motivations, theoretical derivations, and research focus, so it doesn't affect the novelty of our proposed method.** We have a few observations to support our claims:
>
> 1. EMaQ neither mentioned expressive policies (or the MADE architecture) in the Abstract Section nor in the Introduction Section in its manuscript. Policy expressivity is only briefly talked about in Section 3.5 on the methodology. On the contrary, the main focus of our research is to discuss and explain how policy expressivity may affect the performance of algorithms.
>
> 2. The focus of EMaQ's experiments is to demonstrate how the hyperparameter $N$ affects the performance of the algorithm instead of how an expressive policy affects the performance of the algorithm, though they have compared it with a VAE-based baseline to support their architecture choices. In contrast, we aim to visualize the importance of expressive policies by designing new tasks and comparing our method with other baselines as well as other generative models in detail (Figure 1, 3, Table 4, Section 3.1, 6.2, 6.3).
>
> 3. EMaQ actually concentrates on using models with good inductive bias instead of high expressivity, as we quote from its paper: "We do not believe that autoregressive models are intrinsically better than VAEs", and "Our results suggest that for a given domain, focusing efforts on building-in good inductive biases in the generative models and value functions might be sufficient to obtain strong offline RL performance".
>
> 4. EMaQ is derived from BCQ, where behavior policy serves as an implicit constraint of the final policy, while our method aims to address problems in weighted regression (AWAC, CRR, etc.), where the behavior policy is explicit prior of the final policy.
>
> **Q3: Moreover, the value computation technique is similar to a soft version of EMAQ.**
>
> **A3:** The value computation technique used in our method and EMaQ only bear limited similarity, and **the similarity lies only in the empirical implementation part which was not claimed to be a contribution of our method**:
>
> 1. The core contribution of our proposed planning-based in-sample Q-operator is that we combine multi-step Q learning with one-step bootstrapping which provably ensures policy improvement while still guaranteeing a fast-contraction rate. However, EMaQ only considers one-step bootstrapping.
>
> 2. As we have described in the background of EMaQ above, although EMaQ and our method both use N samples to calculate the Q-targets, the $N$ used for EMaQ aims to interpolate between $\mathcal{T}^\mu$ and $\mathcal{T}^*$ while it's only a Monte Carlo sample number for our proposed method. For experiments, the main focus of EMaQ is to study how $N$ affects the property of the backup Q-operator and the performance of the algorithm, while two hyperparameters $N$ and $\alpha$ are fixed to constants for all tasks in our research (N=32, $\alpha=20$). The soft version of EMaQ does not change the theoretical differences between EMaQ and our proposed method. Besides, it is only briefly discussed in Appendix J of EMaQ's manuscript with no further experiments or analysis.
>
> 3. The big empirical performance gap between SfBC and EMaQ in Antmaze tasks could also partly support the novelty and contribution of our proposed implicit in-sample Q-operator because our proposed method helps suppress extrapolation error over long planning horizons.

---

> ### Author Response · Authors · 2022-11-14
> **Response (Part 3/3)**
>
> **Q4: Right now, the main weakness is a lack of clarity. It is not clear how the method is implemented, and the paper can be significantly improved by expanding the implementation details ... it might be difficult to reproduce the results in the paper.**
>
> **A4:** We have made the following adjustments to improve the clarity and ensure the reproducibility of our paper:
>
> 1. We have submitted our source code in the supplementary material to ensure reproducibility.
>
> 2. We have **significantly** expanded the provided implementation details in Appendix B (especially on computational cost and sampling method ).
>
> 3. We have increased independent seeds (3 to 4) for MuJoCo environments and test seeds (20 to 100) for the Antmaze environments.
>
> 4. We have rewritten the pseudocode in Appendix A to include more implementation details.
>
> 5. We have provided ablation studies for two important hyperparameters (value iteration $K$ and diffusion step $D$) and provided training curves of our algorithm.
>
> We hope we can clear the reviewer's concern and clarify our algorithm's implementation. If the reviewer has any further questions or suggestions, we would be very happy to address them.
>
> **Q5: At the end of section 4.2, it's mentioned that the n-step operator proposed in this work works in the stochastic setting. However, this statement is not completely clear. For example, does it mean that it's unbiased?**
>
> **A5:** The statement above does not indicate biased or unbiased estimate.
>
> We made this statement because a prior method VEM [1] was derived under the assumption that the environment dynamics is deterministic ($r(s, a)$ and $P(\cdot| s, a)$ are both **Dirac**) and could not be applied to stochastic environments (A main concern of VEM, raised by one of its reviewers. [Reviewer FzBf, C1-E4](https://openreview.net/forum?id=RCZqv9NXlZ&noteId=gbEGtVfjxo)).  In contrast, our method has no such limitations ($r(s, a)$ and $P(\cdot| s, a)$ could be any distribution instead of just Dirac). **This is actually a claimed technical contribution of our method.**
>
> We thank the reviewer for making us realize that we should have provided more background information to improve clarity. We have added a new section to talk about the connection between VEM and our proposed method in detail in Appendix G.1 (briefly summarized below)
>
> **Additional background of A5:**
>
> A simplified version of $\mathcal{T}_{\text{vem}}$ is defined as:
>
> $\mathcal{T}_{\text{vem}} V(s) := \max _{0 \leq n \leq c}\{(\mathcal{T}^{\mu})^{n}\mathcal{T}_\mu^\tau V(s)\},$
>
> which is is built on $\mathcal{T}_\mu^\tau$, an expectile-based V-learning operator proposed by VEM:
>
> $\mathcal{T}_ \mu^\tau V(s):=\mathbb{E}_{a \sim \mu(\cdot|s)} [ \tau [r(s, a) + \gamma V(s')] + (1-\tau ) V(s)  \quad  \text { if }  r(s, a) + \gamma V(s') \geq V(s) ]$
>
> $\mathcal{T}_ \mu^\tau V(s):=\mathbb{E}_{a \sim \mu(\cdot|s)} [ (1-\tau )  [r(s, a) + \gamma V(s')] + \tau V(s)  \quad  \text { if }  r(s, a) + \gamma V(s') < V(s) ]$
>
> $\tau \in (0, 1)$ is the hyperparameter for expectile operation.
>
> VEM requires comparing $V(s)$ and $r(s,a) + \gamma V(s')$ for the theoretical derivation of the expectile operator $\mathcal{T}_\mu^\tau$, which cannot be applied to stochastic environments both in theory and in implementation. This is because $V(s')$ ($s' \sim P(\cdot| s, a)$) is a random variable in stochastic environments while $V(s)$ is an arbitrary scalar function, which is not comparable. In other words, $P(s'|s,a)$ and $r(s,a)$ have to be Dirac for VEM, but could be any distribution for our proposed method since we require no expectile operation.

---

> ### Author Response · Authors · 2022-11-18
> **Looking forward to further discussions!**
>
> Dear reviewer,
>
> We were wondering if our response and revision have resolved your concerns. In our responses, we focus on discussing the connection and difference between our work and EMaQ and clarifying the implementation of our paper with extensive implementation details and provided source code. If our response has addressed your concerns, we would be very grateful if you could re-evaluate our work.
>
> If you have any additional questions or suggestions, we would be happy to have further discussions.
>
> Best regards,
>
> The Authors

---

> ### Author Response · Authors · 2022-12-01
> **Your feedback is critical to us!**
>
> Dear reviewer,
>
> We were wondering if our response and revision have resolved your concerns. In our responses, we focus on discussing the connection and difference between our work and EMaQ and clarifying the implementation of our paper with extensive implementation details and provided source code. We are really looking forward to discussing this with you so that we could continually improve our work. Your feedback is critical to us.
>
> If our response has addressed your concerns, we would be very grateful if you could re-evaluate our work.
>
> Best regards,
>
> The Authors

---

> > ### Comment · Reviewer_dfM5 · 2022-12-03
> > **Re: Your feedback is critical to us!**
> >
> > Thanks for addressing some of my concerns. I have increased my score to 6.

---

> > > ### Author Response · Authors · 2022-12-04
> > > **Thank you for raising the score! We are happy to address any further concerns!**
> > >
> > > We are grateful that you re-evaluated our work and raised the score.  We are also pleased to see that some of your concerns have been addressed. Please let us know if you still have further concerns or any suggestions! We are more than happy to discuss with you so that we can continually improve our work.
> > >
> > > Thank you!

---

### Official Review · Reviewer_eCnE · 2022-10-24

**Confidence:** 4
**Correctness:** 3
**Technical Novelty And Significance:** 3
**Empirical Novelty And Significance:** 3
**Recommendation:** 6

**Clarity, Quality, Novelty And Reproducibility:**

The paper is well structured and easy to read. The originality is limited. Code is not provided and the paper misses some implementation details, so the reproducibility is limited.

**Strength And Weaknesses:**

Strengths:
1. The performance of SfBC evaluated on D4RL is comparable to the current SOTA methods.
2. The authors demonstrate the importance of using expressive policy class through toy examples.
3. The paper is easy to follow and well written.

Weaknesses:
1. This paper lacks novelty. Policy improvement via weighted regression has been well studied in many papers, such as IQL and AWAC paper. The proposed Q-learning via in-sample planning is motivated from Ma et al. 2022. Applying diffusion models in offline RL is new while using diffusion models for behavior cloning seems an easy plug-in.
2. The paper misses some details about implementation.
3. Computational complexity is a concern and not discussed.


**Summary Of The Paper:**

This paper proposes to learn the offline RL policy by a coupling of pure behavior cloning and importance sampling. The behavior cloning is modeled by a denoising diffusion probabilistic model, and an implicit in-sample Q-learning is applied to estimate the importance sampling weight. Experiments are done in the D4RL benchmark datasets to demonstrate the effectiveness of the method.

**Summary Of The Review:**

This paper proposes to separate the policy learning into two parts: 1) pure behavior cloning via diffusion models; 2) importance sampling via in-sample Q-learning. The idea of using diffusion models as an expressive policy is appealing. However, I do have concerns about the novelty and computational complexity.

As I mentioned in the weaknesses, most of the components from the paper are proposed by prior works. Moreover, by looking at the Algorithm 1, performing implicit in-sample planning has high computational complexity. There are two for loops, one for K and the other for N, which is dataset size and usually huge. Diffusion models are notoriously slow in sampling, while the training algorithm needs samples from diffusion models within the K$\times$N loop, which has too high complexity.
I also have some detailed questions as follows:
1. What is the number of timesteps for the diffusion model used in the paper? I guess an ablation study on the number timesteps is needed to show the effect of it.
2. Better provide a training curve in the appendix to show the training stability.
3. For the reported values in Table 1, what are these values? Are them the statistics based on the last points after the training or the highest points during the training?
5. The policy network is much larger than the current policy networks, which makes the computational cost expensive and comparison not very fair.
6. Expect to an ablation study on the K, since it is a key parameter in Algorithm 1.
7. Minor: typo in Equation (12), I think it is $R^{(k-1)}_{n+1}$.

---

> ### Author Response · Authors · 2022-11-14
> **Response (Part 1/3)**
>
> We thank the reviewer for his very detailed suggestions and really careful reading.
>
> **Q1: This paper lacks novelty. Policy improvement via weighted regression has been well studied in many papers, such as IQL and AWAC paper.**
>
> **A1:** We respectfully disagree with this statement. We think the fact that weighted regression has been widely used and studied in many papers (e.g., IQL, AWAC) actually **strengthened** the novelty and the importance of our proposed method. This is because **our method exactly aims to address existing problems of weighted regression methods instead of being one of them**. We believe there is some misunderstanding here that needs to be clarified. Although our proposed method shares the same theoretical foundation with weighted regression methods, it is not one of them.  one of our main contributions is that we observe existing weighted regression methods require calculating exact log-likelihood which limits policy expressivity. Such limits introduce unnecessary extrapolation error during Q-learning and may have been the bottleneck of existing algorithms (We recently noticed that such a finding was shared by a concurrent work also under the [submission of ICLR, **Diffusion-QL**](https://openreview.net/forum?id=AHvFDPi-FA)). To address this problem, we introduce our method: sampling from behavior candidates in order to adopt a much more expressive generative behavior modeling method (diffusion models). **Our method improves over the existing weighted regression method in terms of policy expressivity and diversity** (As visualized in a diverse-optimal-solution task that we propose in Figure 1 and Figure 3). This is also the key insight into why our method yields strong empirical performance compared with prior methods.
>
> **Q2: Where the novelty of our proposed in-sample planning-based operator lies compared with the Q-learning method proposed by VEM (Ma et al. 2022) ?**
>
> **A2:** The proposed in-sample planning-based Q operator in our paper is by no means a trivial application of VEM [4]. **Our proposed operator is technically novel.** We briefly summarize the novelty of our proposed method, and compare three key differences between VEM and our method:
>
> First, a simplified version of $\mathcal{T}_{\text{vem}}$ is defined as:
>
> $\mathcal{T}_{\text{vem}} V(s) := \max _{0 \leq n \leq c}\{(\mathcal{T}^{\mu})^{n}\mathcal{T}_\mu^\tau V(s)\},$
>
> which is is built on $\mathcal{T}_\mu^\tau$, an expectile-based V-learning operator proposed by VEM:
>
> $\mathcal{T}_ \mu^\tau V(s):=\mathbb{E}_{a \sim \mu(\cdot|s)} [ \tau [r(s, a) + \gamma V(s')] + (1-\tau ) V(s)  \quad  \text { if }  r(s, a) + \gamma V(s') \geq V(s) ]$
>
> $\mathcal{T}_ \mu^\tau V(s):=\mathbb{E}_{a \sim \mu(\cdot|s)} [ (1-\tau )  [r(s, a) + \gamma V(s')] + \tau V(s)  \quad  \text { if }  r(s, a) + \gamma V(s') < V(s) ]$
>
> $\tau \in (0, 1)$ is the hyperparameter for expectile operation.
>
> 1. VEM requires comparing $V(s)$ and $r(s,a) + \gamma V(s')$ for the theoretical derivation of the expectile operator $\mathcal{T}_\mu^\tau$, which cannot be applied to stochastic environments both in theory and in implementation (A main concern of VEM, raised by one of its reviewers. [Reviewer FzBf, C1-E4](https://openreview.net/forum?id=RCZqv9NXlZ&noteId=gbEGtVfjxo)). This is because $V(s')$ ($s' \sim P(\cdot| s, a)$) is a random variable while $V(s)$ is an arbitrary scalar function, which is not comparable. In other words, $P(s'|s,a)$ **and** $r(s,a)$ **have to be Dirac for VEM, but could be any distribution for our proposed method** since we require no expectile operation.
>
> 2.  **VEM is only applicable to algorithms that use expectile regression (e.g., IQL), while our method has no such limitations**. This is because VEM is built on an expectile operator $\mathcal{T}_\mu^\tau$ while ours is built on a more general Q-operator $\mathcal{T}^\pi$.  The theoretical properties of the two operators are also different (See our proofs in Proposition 1 in Appendix C).
>
> 3. VEM is a V-operator that learns a value function, while our method introduces a Q-operator which learns a Q function.
>
> We notice that this part was not discussed in detail in the previous manuscript due to the page limit, and may cause some misunderstanding. We have strengthened this in the updated paper by adding a new section (Appendix G) in the updated paper to discuss the connection and the difference between the two methods in more detail.
>
> [4] Offline Reinforcement Learning with Value-based Episodic Memory. https://arxiv.org/abs/2110.09796

---

> ### Author Response · Authors · 2022-11-14
> **Response (Part 2/3)**
>
> **Q3: Applying diffusion models in offline RL is new while using diffusion models for behavior cloning seems an easy plug-in.**
>
> **A3:** Technically, applying diffusion models for behavior cloning is straightforward given that we need a much more expressive generative modeling method, but we believe this doesn't affect the empirical contributions of the paper. Diffusion models have not been previously used to model the behavior policy in reinforcement learning so there is few prior work that we could reference. As a result, we have made much effort trying to design an appropriate network architecture, to search for suitable hyperparameters, and to compare different sampling methods with a large number of experiments. We have also made many efforts trying to reduce the training and inference time required (which will be discussed in detail later) to make the algorithm more practical.
>
> Besides, we believe keeping our method simple can directly reveal the fact that policy expressivity is an important factor in offline RL (The key finding in this paper).
>
> **Q4: Moreover, by looking at Algorithm 1, performing implicit in-sample planning has high computational complexity. There are two for loops, one for K and the other for N, which is dataset size and usually huge.**
>
> **A4:** We think this runtime cost is totally acceptable since the ``two for loops'' part mentioned above takes only about 6.3*($K \in [2,5]$) minutes. In more detail, the "two for loops'' refers to the update-value-target runtime, which is a part of the critic training. Training the critic (action evaluation) model is roughly 155 minutes for the "halfcheetah-medium-v2'' task (1M data points) if we set value iteration K=5, so a total 122k gradient steps are considered. For comparison, CQL roughly requires 4 hours of training for the same task with the same hardware setup. We think the main understanding gap here is due to three reasons:
>
> 1. K in the outer loop is actually a very small number (at max 5 for all tasks).
>
> 2. The inner loop for N is actually not executed sequentially, but parallelized with a batchsize of 4096 to utilize the computing power of GPUs.
>
> 3. We require much fewer gradient steps compared with CQL (122k v.s 1M). **This is one of the claimed contributions or benefits of our proposed method (fast contraction rate)**.
>
> We notice that the previous pseudocode may cause such an understanding gap. We thus thank the reviewer for the comment and have updated the pseudocode (Appendix A).
>
> **Q5: What is the number of timesteps for the diffusion model used in the paper? I guess an ablation study on the number timesteps is needed to show the effect of it.**
>
> **A5:** We adopt recent advances in diffusion sampling technique [1] and use **a uniform 15 diffusion steps** for the final performance of the paper. Generally, we find that 10-25 diffusion steps yield similar performance, while a diffusion step of 5 would result in drastic performance degrade. We have provided an ablation study to compare the effect of using different diffusion steps in Table 2 and Figure 8 (summarized below).
>
> | Diffusion Steps D                          |  5 steps | 10 steps  | 15 steps | 25 steps |
> |--------------------------------------------|----------|-----------|----------|----------|
> | Averaged Performance (Locomotion)          | 2.3      | 72.9      | **75.6**    | 74.4     |
> | Averaged Performance (Antmaze)             | 5.5      | 65.7      | **74.2**    | 73.0     |
> | Averaged Runtime (1 episode, \# envs=1)    | 22.3 s   | 38.0 s    | **50.0 s**  | 93.0 s   |
> | Averaged Runtime (1 episode, \# envs=20) | 1.5 s    | 2.5 s     | **3.2 s**  | 5.0 s    |
>
>
> Specifically, we make the following efforts in the updated manuscript to reduce the averaged evaluation runtime and make our algorithm practical:
>
> 1. We adopt a recent advance in diffusion-specialized ODE solver [1] to replace the standard 4th order Runge–Kutta solver and reduce D from typically 35-50 to 10-15 steps in our work.
>
> 2. We implement a parallel evaluation scheme [5] that could help evaluate the algorithm under multiple test seeds at the same time, allowing us to significantly reduce the evaluation runtime by utilizing the parallel computing power of GPUs.
>
> [1] DPM-Solver: A Fast ODE Solver for Diffusion Probabilistic Model Sampling in Around 10 Steps https://arxiv.org/abs/2206.00927
>
> [5] Efficient parallel methods for deep reinforcement learning https://arxiv.org/abs/1705.04862

---

> ### Author Response · Authors · 2022-11-14
> **Response (Part 3/3)**
>
> **Q6: Computational complexity is a concern and not discussed.**
>
> **A6:** The training runtime roughly matches existing algorithms as we have discussed in **A4**, so the computational complexity is acceptable. For sampling (evaluation), as is discussed in **A5**, we managed to reduce the evaluating runtime to 3.2s per episode (1k action steps) when using 20 parallel environments and **a diffusion step of 15**. This is about the same time required for a non-diffusion algorithm that doesn't use parallel evaluation techniques, making our algorithm practical.
>
> We have expanded experiment details to provide more experiments of runtime required both for algorithm training and evaluation under different diffusion steps and parallel environment numbers (Table 2 and Figure 7 Appendix B.1). Hopefully, this could address the reviewer's concern.
>
> **Q7: Better provide a training curve in the appendix to show the training stability.**
>
> **A7:** We have provided the training curves under four different diffusion steps for MuJoCo and Antmaze tasks in Appendix F.
>
> **Q8: Expect to an ablation study on the K, since it is a key parameter in Algorithm 1.**
>
> **A8:** We have provided an ablation study on $K \in \{1,2,3,4,5\}$ in Figure 4. In general, we empirically find that MuJoCo tasks are less sensitive to the value iteration $K$, so we set $K=2$ for all MuJoCo tasks. In contrast, a larger $K = 5$ significantly improves the performance in Antmaze tasks, indicating Antmaze tasks require stable dynamic programming over long horizons, also demonstrating the effectiveness of our method.
>
> **Q9: For the reported values in Table 1, what are these values? Are them the statistics based on the last points after the training or the highest points during the training?**
>
> **A9:** They are statistics based on the last points.
>
> **Q10: The policy network is much larger than the current policy networks, which makes the computational cost expensive and comparison not very fair.**
>
> **A10:** We believe the comparison with referenced baselines is fair for the following reasons:
>
> 1. SfBC (our method) requires using a more expressive policy, this partly means a network of high capacity and more parameters, which we think is a trade-off between policy expressivity and computational cost.
>
> 2. We included two baselines that also use high-capacity networks for comparison. Concretely, DT (NIPS 2021, [6]) uses transformers which has 0.6-1.5M parameters; Diffuser (ICML 2022 Oral, [7]) uses an architecture that combines diffusion models with the transformer architecture which has 4M parameters. The parameters used in SfBC are approximately 1.6M for all tasks. However, the FLOPs required by our method is more than **10 times less** than these sequential-based methods because we do not operate at the trajectory level.
>
> 3. Other referenced baselines which use a smaller policy network are not likely to have any performance improvement if switching to a high-capacity model. In other words, their reported results have mostly been finetuned. We have provided experiment results for IQL (our best-performing baseline) below to support our claims.
>
>
> |                     | Half-MR      | Hopper-MR   | Walker-MR     | Half-M        | Hopper-M    |
> |---------------------|--------------|-------------|---------------|---------------|-------------|
> | IQL (small network) | 43.8±0.5     | 77.3±11.4   | 76.0±1.0      | 47.6±0.0      | 66.2±2.4    |
> | IQL (large network) | 41.1±0.3     | 60.4±0.5    | 62.0±6.1      | 48.0±0.0      | 60.3±0.5    |
> |                     |              |             |               |               |             |
> |                     | **Walker-M** | **Half-ME** | **Hopper-ME** | **Walker-ME** | **Average** |
> | IQL (small network) | 79.7±1.1     | 87.3±1.7    | 84.1±26.4     | 109.9±0.2     | **74.7**    |
> | IQL (large network) | 79.3±0.6     | 94.5±0.1    | 94.7±22.0     | 110.1±0.3     | **72.3**    |
>
> (results reported under 3 random seeds, 20 test seeds, large network refers to the one similar to the model used in SfBC, small network is what used in the original paper)
>
> **Q11: Minor: typo in Equation (12), I think it is $R_{n+1}^{k-1}$**
>
> **A11:** We sincerely thank the reviewer for the careful reading. The typo is fixed now, though we think the superscript should be $k$ instead of $k-1$.

---

> ### Author Response · Authors · 2022-11-18
> **Looking forward to further discussions!**
>
> Dear reviewer,
>
> We were wondering if our response and revision have resolved your concerns. In our responses, we focus on clarifying the novelty of our proposed method and providing extensive implementation details.  If our response has addressed your concerns, we would be grateful if you could re-evaluate our work.
>
> If you have any additional questions or suggestions, we would be happy to have further discussions.
>
> Best regards,
>
> The Authors

---

> ### Author Response · Authors · 2022-12-01
> **Your feedback is critical to us!**
>
> Dear reviewer,
>
> We were wondering if our response and revision have resolved your concerns. In our responses, we focus on clarifying the novelty of our proposed method and providing extensive implementation details (especially on computational complexity and runtime). We are really looking forward to discussing this with you so that we could continually improve our work. Your feedback is critical to us.
>
> If our response has addressed your concerns, we would be very grateful if you could re-evaluate our work.
>
> Best regards,
>
> The Authors

---

> > ### Comment · Reviewer_eCnE · 2022-12-04
> > **Thank for the response**
> >
> > First, sorry for the late reply. I thank the authors for the detailed response, which addresses my main concerns on the computational cost and missing implementation details. I have increased my rating.

---

> > > ### Author Response · Authors · 2022-12-05
> > > **Thank you! Do share with us if you have any further concerns!**
> > >
> > > We are pleased to see that we can address your main concerns and that you raised the rating of our work. Please let us know if you still have further concerns or any suggestions! We are more than happy to discuss with you so that we can improve our work.
> > >
> > > Thank you!

---

### Official Review · Reviewer_3dPg · 2022-10-25

**Confidence:** 4
**Correctness:** 4
**Technical Novelty And Significance:** 4
**Empirical Novelty And Significance:** 4
**Recommendation:** 6

**Clarity, Quality, Novelty And Reproducibility:**

Clarity: This paper is fairly well written.

Quality: The paper is of high quality.

Novelty: The paper applies diffusion models to offline RL and uses a well-known solution to solve for high value actions while minimizing KL divergence from the behavior policy. The only other work applying diffusion models to model-free RL is [1], which was presumably contemporaneous with this work and the specifics of the two approaches are distinct.

Reproducibility: The source code is not provided (it is suggested that it will be made available later). For this style of paper, with benchmarks on a standardized set of tasks in a rapidly area of research this seems below the norm. In particular, I would imagine people would struggle to reproduce these results without access to the code.

[Updated during rebuttal. The authors now provide source code. I did not test it but this addresses my concerns about reproducibility].

**Strength And Weaknesses:**

Strengths:
1. The paper is well written and explains the method.

2. The method is well motivated.

3. The empirical results show good.

4. The topic is off significant interest.

Weaknesses:
1. I have concerns about the reproducibility, I believe the norm (particularly for this topic) is to include the source code in the submission rather than a promise of future code. Generally, particularly in RL, people have struggled to exactly reproduce results just from pseudocode.

For example, I did not see in Appendix C the number of diffusion steps used (quite an important detail). It would also be helpful to discuss the runtime costs of sampling from diffusion models.

Ideally the results would be over more than 3 seeds, but this is not vital.

Other:

[1] Is a closely related work that uses diffusion models for offline RL that should be mentioned in related work (although the method is not identical). It also makes similar claims about the importance of expressive, multi-modal policies.

[1] Wang, Zhendong, Jonathan J. Hunt, and Mingyuan Zhou. "Diffusion policies as an expressive policy class for offline reinforcement learning." arXiv preprint arXiv:2208.06193 (2022).


**Summary Of The Paper:**

This work uses a diffusion model as an expressive generative model for behavior cloning. They then turn this into a offline RL algorithm with model improvement by learning a Q-function and performing a form of importance sampling to rejection sample action samples with high value by sampling from the behavior clone model and resampling based on Q. They test this on some standard offline RL benchmarks.


**Summary Of The Review:**

A well-present piece of work applying diffusion modeling to offline RL. It would be strengthened by making the source code available to ensure it can be reproduced and built on [I note this concern is fixable during the rebuttal period].

[Updated]

The authors have uploaded source code, cited additional work and made other improvements. I have increase my rating.

---

> ### Author Response · Authors · 2022-11-14
> **Response**
>
> We thank the reviewer for his expertise and valuable suggestions and feel encouraged by the reviewer's comment.
>
> **Q1: I believe the norm is to include the source code in the submission rather than a promise of future code.**
>
> **A1:** We thank the reviewer for this suggestion. We have submitted the source code in the supplementary material to ensure reproducibility.
>
> **Q2: Ideally the results would be over more than 3 seeds, but this is not vital.**
>
> **A2:** Three random seeds is a shared evaluation metric adopted by many of our referenced baselines (D4RL, Diffuser, DT, etc.). Still, we have increased the random seeds used from 3 to 4 for MuJoCo and Antmaze tasks in order to clear the reviewer's concern. We have also increased the test seed number for all Antmaze tasks from 20 to 100 to remove possible randomness. After the update, the average performance for MuJoCo tasks almost doesn't change. The average performance for Antmaze tasks drop by 4 percent but still remains the highest among all referenced baselines. Note that the performance numbers for Kitchen and Maze2d have not been updated due to the very limited time for rebuttal revision. We plan to update those numbers before the camera-ready version of the manuscript.
>
> **Q3: I did not see in Appendix C the number of diffusion steps used (quite an important detail).**
>
> **A3:** In the previous version of the paper, we use a standard Runge–Kutta ODE solver which requires 35 to 50 diffusion steps given different environments. **In the updated manuscript, we are able to decrease the diffusion steps required to 15 steps in all tasks.** This is because we adopt a recent advance in specialized diffusion sampler, namely DPM-solver [1]. We additionally conduct a series of experiments to study the influence of diffusion steps on the overall performance (Table 2, Appendix B.1, summarized below). Generally speaking, we find that 10-25 diffusion steps result in similar overall performance.
>
> [1] DPM-Solver: A Fast ODE Solver for Diffusion Probabilistic Model Sampling in Around 10 Steps. https://arxiv.org/abs/2206.00927
>
> **Q4: It would also be helpful to discuss the runtime costs of sampling from diffusion models.**
>
> **A4:** For a concrete example, it takes 3.2s to evaluate our algorithm for every episode (1000 steps) in the  "halfcheetah-medium-v2" task, when using 15 diffusion steps and 20 parallel environments (Hardware being a 2080Ti GPU). This runtime cost roughly matches the runtime of non-diffusion algorithms (diffusion step is 1) that don't use parallel evaluation. Although sampling from diffusion models is known to be slow, we use two techniques to make our method more experimentally practical.
>
> 1. We leverage recent advances in the diffusion sampling method, which allows us to sample from the diffusion model with only 15 steps.
>
> 2. We implement a parallel evaluation scheme similar to [5]. Instead of sampling actions sequentially for multiple test seeds, we sample all actions in a single network inference by using a larger batchsize. This allows us to fully utilize the parral computing power of GPUs and greatly accelerates action sampling.
>
> We have conducted an ablation study to discuss the runtime costs of diffusion sampling, the detailed results are provided in Figure 7 and Table 2 in Appendix B.1.
>
> [5] Efficient parallel methods for deep reinforcement learning. https://arxiv.org/abs/1705.04862
>
> **Q5: Diffusion-QL Is a closely related work that uses diffusion models for offline RL that should be mentioned in related work (although the method is not identical).**
>
> **A5:** We have mentioned and cited this work (Diffusion-QL) in Section 4. Diffusion-QL [3] and SfBC (our method) are two **concurrent and independent** lines of work. We have noticed that their work is also under [submission to ICLR 2023](https://openreview.net/forum?id=AHvFDPi-FA).
>
> In principle, it is unnecessary for us to compare to a concurrent work. Still, we list three aspects from which the two works are easily distinguishable to assist in understanding and clarifying our method:
>
> 1. SfBC is theoretically derived from weighted regression methods where the behavior policy serves as an explicit prior of the final policy (e.g. AWAC, CRR), while Diffusion-QL is based on another line of methods where the behavior policy serves as an implicit regularization of the final policy (e.g. TD3+BC, BCQ).
>
> 2. SfBC additionally proposes a computationally efficient in-sample planning Q-operator to further avoid selecting OOD actions, while Diffusion-QL uses one-step bootstrapping.
>
> 3. SfBC samples from the learned policy using a rejection sampling technique, while diffusion-QL uses an approximated guided sampling method. Besides, the implementation details of the two algorithms are significantly different.
>
> [3] Diffusion-QL: Diffusion Policies as an Expressive Policy Class for Offline Reinforcement Learning. https://arxiv.org/abs/2208.06193

---

> > ### Author Response · Authors · 2022-11-14
> > **Extra information**
> >
> > **Summarized results for ablation study of diffusion step D. (4 independent seeds + 20-100 test seeds)**
> >
> > | Diffusion Steps D                          |  5 steps | 10 steps  | 15 steps | 25 steps |
> > |--------------------------------------------|----------|-----------|----------|----------|
> > | Averaged Performance (Locomotion)          | 2.3      | 72.9      | **75.6**    | 74.4     |
> > | Averaged Performance (Antmaze)             | 5.5      | 65.7      | **74.2**    | 73.0     |
> > | Averaged Runtime (1 episode, \# envs=1)    | 22.3 s   | 38.0 s    | **50.0 s**  | 93.0 s   |
> > | Averaged Runtime (1 episode, \# envs=20) | 1.5 s    | 2.5 s     | **3.2 s** | 5.0 s    |
> >
> > Detailed results and training curves can be found in Appendix B.1 and Appendix F.

---

> ### Author Response · Authors · 2022-11-18
> **Looking forward to further discussions!**
>
> Dear reviewer,
>
> We were wondering if our response and revision have resolved your concerns. In our responses, we focus on ensuring the reproducibility of our work, including providing the source code, increasing random seeds, and providing more ablation studies. We have also improved over the prior method to reduce the sampling time and provided a thorough ablation study of diffusion steps. If our response has addressed your concerns, we would be grateful if you could re-evaluate our work.
>
> If you have any additional questions or suggestions, we would be happy to have further discussions.
>
> Best regards,
>
> The Authors

---

> > ### Comment · Reviewer_3dPg · 2022-11-23
> > **Thanks**
> >
> > Thanks. I have increased my rating.

---

> > > ### Author Response · Authors · 2022-11-26
> > > **Thank you for raising the scores**
> > >
> > > We are happy to see that we could address the reviewer's concern. We would like to thank the reviewer for raising the score.

---

> > > ### Author Response · Authors · 2022-12-05
> > > **Please share with us if you have any additional concerns!**
> > >
> > > Dear reviewer,
> > >
> > > Thank you again for raising the rating of our work a few days before. We feel very encouraged by your review, praising our work as "of significant interest", "well motivated" and "well-presented". We find your main concern about our work is about reproducibility, which we addressed by providing our source code,  increasing random seeds, and expanding implementation details.
> > >
> > > As a result, we were wondering what could we do to further address your concerns for a higher rating. Our very first guess on the reason that prevents you from a higher rating is that you might also share the concerns raised by two other reviewers. We would like to raise your attention that reviewers dfM5 and eCnE both replied to us very recently, proposed no further concerns, and increased the rating. We thus hope our responses to them could also solve your potential concerns.
> > >
> > > At the same time, we will be extremely happy if you still have **any** additional concerns to share with us. Your opinion is critical to us. We will try our best to address your concerns.
> > >
> > > Best regards,
> > >
> > > The Authors

---

### Author Response · Authors · 2022-11-14
**Revision summary**

We want to thank all the reviewers for their detailed comments. We summarize all the reviewers' concerns into three categories: 1. On the experimental details and the reproducibility of the proposed algorithm; 2. On the runtime cost and computational complexity of the algorithm.  3. On the algorithm's connection with several related works.

We have made a number of changes to address reviewers' suggestions and concerns. A short summary of the modifications made:
1. Provide the source code of our paper in the supplementary material to ensure reproducibility.
2. Significantly expand the implementation details in Appendix B to provide more information on the implementation of our method (especially on the runtime and the sampling method).
3. Increase the test seeds for Antmaze tasks from 20 to 100. Increase the independent experiment seed numbers from 3 to 4 for Antmaze and MuJoCo.
4. Reduce diffusion steps required by three times (now 15 steps) by adopting a recent advance in diffusion-specialized ODE solver [1] to replace the standard Runge–Kutta solver for algorithm evaluation.
5. Conduct an ablation study on the diffusion steps used.
6. Conduct an ablation study on the value iteration K.
7. Add citations of several related works [2,3], and add a new section (Appendix G) to discuss the connection and difference between our method and two related methods (VEM [4] and EMaQ[2]).
8. Provide training curves of the algorithm to demonstrate the stability.
9. Fix several typos in the paper.


[1] DPM-Solver: A Fast ODE Solver for Diffusion Probabilistic Model Sampling in Around 10 Steps. https://arxiv.org/abs/2206.00927

[2] EMaQ: Expected-Max Q-Learning Operator for Simple Yet Effective Offline and Online RL. https://arxiv.org/abs/2007.11091

[3] Diffusion Policies as an Expressive Policy Class for Offline Reinforcement Learning. **Concurrent work**, submission to ICLR 2023: https://openreview.net/forum?id=AHvFDPi-FA.

[4] Offline Reinforcement Learning with Value-based Episodic Memory. https://arxiv.org/abs/2110.09796

---

> ### Author Response · Authors · 2022-11-23
> **Looking forward to further discussions!**
>
> Dear reviewers,
>
> We were wondering if our responses and revision have resolved your concerns. In our responses, we have added experiment results based on your suggestions and clarified several claims in the paper. Please let us know if we have addressed your concerns. We are more than delighted to have further discussions and improve our manuscript.
>
> Best regards,
>
> The Authors

---

### Decision · Program_Chairs · 2023-01-20

**Decision:**

Accept: poster

**Justification For Why Not Higher Score:**

The novelty is limited.

**Justification For Why Not Lower Score:**

The paper is generally well-executed with clear motivation and nice results on a number of challenging offline RL problems, including ant-maze and kitchen. The author response also helped address some reviewer concerns, i.e. around computational complexity and implementation details.

**Metareview: Summary, Strengths And Weaknesses:**

This paper is somewhat borderline, since the novelty of the method is somewhat limited. That said, the paper is generally well-executed with clear motivation and nice results on a number of challenging offline RL problems, including ant-maze and kitchen. The author response also helped address some reviewer concerns, i.e. around computational complexity and implementation details. Thus, I recommend accept.

There are a couple other related works that I think should be included, namely PLAS (https://arxiv.org/abs/2011.07213) and LAPO (https://openreview.net/forum?id=pHd0v8W30O). I'd encourage the authors to incorporate these into the related work discussion in the revised version.


**Note From Pc:**

if the above contains the word "oral" or "spotlight" please see: "oral" presentation means -> notable-top-5% and "spotlight" means -> notable-top-25%. As stated in our emails, we are disassociating presentation type from AC recommendations